# Release of human cytomegalovirus from latency by a KAP1/TRIM28 phosphorylation switch

**Benjamin Rauwel, Suk Min Jang, Marco Cassano, Adamandia Kapopoulou, Isabelle Barde, Didier Trono\***

School of Life Sciences, Ecole Polytechnique Fédérale de Lausanne, Lausanne, Switzerland

**Abstract** Human cytomegalovirus (HCMV) is a highly prevalent pathogen that induces life-long infections notably through the establishment of latency in hematopoietic stem cells (HSC). Bouts of reactivation are normally controlled by the immune system, but can be fatal in immuno-compromised individuals such as organ transplant recipients. Here, we reveal that HCMV latency in human CD34+ HSC reflects the recruitment on the viral genome of KAP1, a master co-repressor, together with HP1 and the SETDB1 histone methyltransferase, which results in transcriptional silencing. During lytic infection, KAP1 is still associated with the viral genome, but its heterochromatin-inducing activity is suppressed by mTOR-mediated phosphorylation. Correspondingly, HCMV can be forced out of latency by KAP1 knockdown or pharmacological induction of KAP1 phosphorylation, and this process can be potentiated by activating NFkB with TNF-α. These results suggest new approaches both to curtail CMV infection and to purge the virus from organ transplants.

## Introduction

Human cytomegalovirus (HCMV), a member of the β-herpes virus family, is a highly prevalent pathogen that induces life-long infections notably through the establishment of latency in hematopoietic stem cells (HSC) (*Sinclair, 2008*). Bouts of reactivation are normally asymptomatic because rapidly controlled by the immune system, but can be fatal in immuno-compromised individuals such as AIDS patients and transplant recipients, in particular when grafts from HCMV-positive donors are given to HCMV-negative individuals (*Sissons and Carmichael, 2002*). Furthermore, primary HCMV infection during pregnancy is a leading cause of congenital malformations of the central nervous system (*Britt, 2008*).

The approximately 250 kb genome of HCMV encodes several hundred proteins, some 14 miRNAs and a few long-noncoding RNAs (*Stern-Ginossar et al., 2012*). Infection of permissive targets such as epithelial cells or fibroblasts leads to a lytic cycle, with a highly orderly transcriptional cascade that first expresses the viral immediate early (IE) genes, the products of which set the cellular stage for the virus and activate the viral early (E) genes. These yield the effectors of viral genome replication, before proteins encoded by the viral late (L) genes finally trigger the formation of new particles. In contrast, when HCMV infects a HSC, the expression of lytic genes is rapidly suppressed and only a few latency-associated transcripts are detected (*Goodrum et al., 2012*). The viral genome is then maintained as a stable episome, without replicating or producing new virions (*Goodrum et al., 2004*). When HCMV-infected HSC are differentiated in vitro into dendritic cells (DCs) and these are induced to mature, the virus resumes a lytic cycle. Accordingly, blood CD14+ monocytes, which are circulating precursors of DCs, harbor latent HCMV.

HCMV reactivation is thus tightly coupled with myeloid differentiation. However, the molecular mechanisms of its persistence in latently infected cells are incompletely understood, albeit known to

**\*For correspondence:** didier. trono@epfl.ch

**Competing interests:** The authors declare that no competing interests exist.

**eLife digest** Human cytomegalovirus (HCMV) is an extremely common virus that causes life-long infections in humans. Most individuals are exposed to HCMV during childhood, and the infection rarely causes any symptoms of disease in healthy individuals. However, in people with weaker immune systems—for example, newborn babies, people with AIDS, or individuals who have received an organ transplant—HCMV can cause life-threatening illnesses.

It is difficult for the immune system to fight the infection because HCMV is able to hide in cells within the bone marrow called hematopoietic stem cells. Inside these cells, the virus can survive in a 'dormant' state for many years, before being reactivated and starting to multiply again. In most people, the immune system manages to control this new outbreak of HCMV, and the virus becomes dormant again, but reactivation of the virus in individuals with weakened immune systems is much more likely to cause serious illness.

The results of previous studies suggest that when HCMV infects the hematopoietic stem cells, human proteins switch off the expression of many virus genes, which makes the virus inactive. The virus can be reactivated when infected stem cells change into a type of immune cell called dendritic cells, but it is not clear how this is controlled.

Here, Rauwel et al. reveal that a human protein called KAP1 is responsible for switching off the virus genes in the stem cells. It does so by interacting with two other proteins to alter the structure of the DNA in these genes. However, if the stem cells are stimulated to change into dendritic cells, KAP1 becomes inactive, which allows the virus genes to be switched on.

Rauwel et al. also show that it is possible to force HCMV out of its dormant state by using drugs to block the activity of KAP1. This may aid the development of treatments that prevent the virus from causing serious illness in patients with weakened immune systems. For example, it could be used to remove dormant HCMV infections from bone marrow before it is transplanted into a new individual.

be associated with epigenetic modifications of the viral chromatin (*Avdic et al., 2011*; *Umashankar et al., 2011*; *Mason et al., 2012*; *Petrucelli et al., 2012*). In latently infected $CD34^+$ HSC or circulating monocytes isolated from seropositive individuals, chromatin at the major immediate early promoter (MIEP) bears histone 3 trimethylated on lysine 9 (H3K9me3), a repressive mark, and heterochromatin protein 1 (HP1) (*Sinclair, 2010*; *Reeves and Sinclair, 2013*). Upon differentiation of latently infected precursors into dendritic cells, H3K9me3 is replaced by high levels of histone acetylation, HP1 is lost, and a lytic cycle is triggered (*Taylor-Wiedeman et al., 1994*; *Mendelson et al., 1996*; *Hahn et al., 1998*; *Reeves et al., 2005*).

KAP1 (KRAB-associated protein 1), also known as TRIM28 (tripartite motif protein 28) or TIF1β (transcription intermediary factor 1 beta) is a transcriptional co-repressor essential for the early embryonic silencing of endogenous retroelements (*Rowe et al., 2010*; *Castro-Diaz et al., 2014*; *Turelli et al., 2014*) and involved in regulating multiple aspects of mammalian homeostasis, including in the hematopoietic system (*Jakobsson et al., 2008*; *Bojkowska et al., 2012*; *Chikuma et al., 2012*; *Santoni de Sio et al., 2012a*, *2012b*; *Barde et al., 2013*). KAP1 binds to the Krüppel-associated box (KRAB) domain present at the N-terminus of KRAB-containing zinc finger proteins (KRAB-ZFPs) (*Friedman et al., 1996*). These constitute the single largest family of transcriptional repressors encoded by the genomes of higher organisms, with close to four-hundred members in either human or mouse, and are endowed with sequence-specific DNA binding ability via a C-terminal array of zinc fingers (*Urrutia, 2003*). KAP1 harbors from its N- to its C-terminus a RING finger, B-boxes, a coiled-coil region, a HP1-binding motif, a PHD finger and a bromodomain (*Iyengar and Farnham, 2011*). The first three of these motifs define the so-called RBCC or TRIM (tripartite motif) region, which is both necessary and sufficient for homo-oligomerization and direct binding to KRAB. The C-terminal effector end of the protein recognizes the backbone of histone tails, and interacts with two histone-modifying enzymes: Mi2α, an isoform of the Mi2 protein found in the NuRD (nucleosome remodeling and histone deacetylation) complex (*Schultz et al., 2001*), and SETDB1 (SET domain, bifurcated 1), an H3K9me3-specific histone methyltransferase (*Schultz et al., 2002*). The H3K9me3 mark in turn creates high affinity genomic binding sites for HP1, bringing more KAP1 complex, which likely explains that KAP1-induced heterochromatin formation can spread several tens of kilobases away from an initial

KAP1 docking site (*Groner et al., 2010*). SETDB1 recruitment is stimulated by sumoylation of the KAP1 bromodomain, the last step of which is mediated intra-molecularly by an E3 ligase activity contained in the PHD domain (*Ivanov et al., 2007*). Within the context of DNA damage, the ATM (ataxia telangiectasia mutated) kinase phosphorylates KAP1 at position 824, resulting in decreased KAP1 sumoylation and SETDB1 recruitment, with secondary loss of KAP1 repressor activity (*White et al., 2006*, *2012*; *Noon et al., 2010*).

Histone 3 lysine 9 trimethylation and HP1 recruitment are thus signatures of KAP1 action. Furthermore, we previously demonstrated that KRAB/KAP-mediated repression is functional within the context of episomal DNA (*Barde et al., 2009*). Finally, the KRAB/KAP pathway has been found to influence the replication of two members of the herpes virus family, Epstein–Barr virus (EBV) (*Liao et al., 2005*) and Kaposi's sarcoma-associated herpes virus (KSHV) (*Chang et al., 2009*; *Cai et al., 2013*). Based on these premises, we investigated a possible role for KAP1 in the control of HCMV latency. We found that KAP1 is a key mediator of this process, and that it is targeted by an mTOR-mediated phosphorylation switch that can be exploited to force the virus out of latency. Our results suggest new avenues not only for controlling HCMV infection but also for purging the virus from organ transplants.

## Results

### KAP1 is necessary for both establishment and maintenance of HCMV latency in HSC

We first infected either MRC5 fibroblasts or cord blood-derived CD34$^+$ cells with the TB40-E HCMV clinical strain. Monitoring the levels of viral DNA and various RNA transcripts during the following days confirmed that lytic replication occurred in MRC5 cells, known to be fully permissive, whereas latency was established in HSC after an initial outburst of viral gene expression (*Figure 1—figure supplement 1A,B*), as previously reported (*Goodrum et al., 2007*). To examine the possible impact of KAP1 on this process, we depleted the corepressor by lentivector-mediated RNA interference, complementing *Kap1* knockdown cells by transduction with a vector expressing an shRNA-resistant *Kap1* allele when relevant. We then infected these cells with TB40-E HCMV. In MRC5 cells, where KAP1 depletion was around 98% (*Figure 1—figure supplement 2A*), levels of viral transcripts indicative of a lytic cycle (the IE genes UL123, UL122, the E gene UL54 and the L gene UL94) were comparable in control and KAP1-depleted cells (*Figure 1A*) and production of viral proteins IE1 and IE2 was unchanged (*Figure 1—figure supplement 2B*), indicating that the regulator is not essential for productive HCMV replication. In unsorted population of HSC exposed to the knockdown lentivector, KAP1 reduction was only 85% but in sorted CD34$^+$ GFP-positive, that is, transduced cells, it reached 95% (*Figure 1—figure supplement 2C,D*). In this KAP1-depleted subpopulation, expression of immediate early, early and late viral genes at 7 days post-infection was increased between 10- and 35-fold compared with control cells (*Figure 1B*), and massive amounts of infectious HCMV particles were released in the supernatant (*Figure 1C*). Furthermore, complementation of KAP1-depleted cells with an shRNA-resistant *Kap1* allele restored HCMV latency (*Figure 1—figure supplement 2E–I*). These results thus indicated that KAP1 is necessary for the establishment of HCMV latency in HSC. We then inverted the sequence of our manipulations to induce *Kap1* knockdown in CD34$^+$ cells that had been infected 7 days earlier with the TB40-E virus (*Figure 1—figure supplement 2J*). 7 days after transduction, we measured the expression of the UL122, UL123, UL54 and UL94 mRNAs, which are found only during a lytic cycle (*Reeves and Sinclair, 2013*) (*Figure 1D*). All transcripts were significantly upregulated in knockdown cells, which accordingly released infectious viral particles (*Figure 1E*). Furthermore, the NF-κB activator TNFα increased viral gene expression and virion production from *Kap1* knockdown but not from control TB40-E-infected CD34$^+$ cells (*Figure 1D,E*). These data indicate that (i) KAP1 is necessary both for the establishment and for the maintenance of HCMV latency in human HSC, and (ii) the relief of KAP1-mediated repression is not in itself sufficient for full HCMV reactivation in CD34$^+$ cells, but provides the ground for stimulation of viral gene expression by HCMV activators such as NF-κB.

### HCMV latency in HSC correlates with KAP1-mediated recruitment of SETDB1 and HP1α and with H3K9 trimethylation

Cells in which HCMV expression was induced following KAP1 depletion kept expressing the CD34 stem cell marker, indicating that viral activation was not simply due to their differentiation into normally permissive cells such as activated DCs or macrophages. Still, as KAP1 exerts pleomorphic

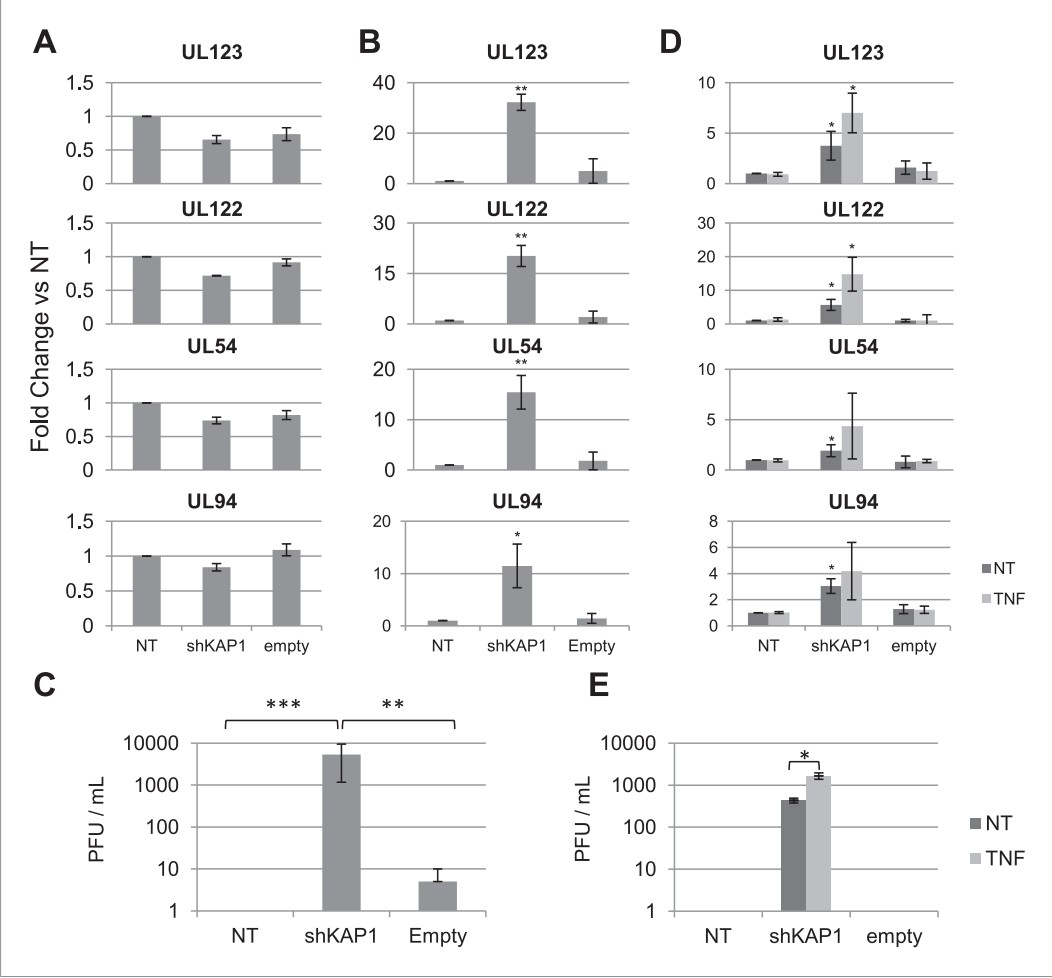

**Figure 1**. KAP1 is required for HCMV latency. RT-qPCR analysis of indicated HCMV genes expression in MRC-5 (**A**) or cord blood CD34+ (**B**, **C**, **D**, **E**) cells infected with the TB40-E strain 3 days after (**A**, **B**, **C**) or 7 days prior to (**D**, **E**) being transduced or not (NT) with lentivectors expressing (shKAP1) or not (empty) a small hairpin RNA against *Kap1*, sorting CD34+ cells for GFP and CD34 expression. RT-qPCRs were performed 7 days after HCMV infection (**A** and **B**) or 7 days after lentiviral transduction (**D** and **E**). Results are presented as average of fold change expression vs NT after GAPDH and β-2M normalization (n = 4, *p < 0.05, **p < 0.01, error bars as s.d.). (**C** and **E**) HCMV production in CD34+ cells was quantified by plaque assay on MRC-5 cells. Results are presented as average of PFU/ml (n = 3, *p < 0.05, **p < 0.01, ***p < 0.001, error bars as s.d.). See *Figure 1—figure supplements 3, 4* for all individual experiments.

The following figure supplements are available for figure 1:

**Figure supplement 1**. KAP1 is necessary for both establishment and maintenance of HCMV latency in HSC.

**Figure supplement 2**. KAP1 is necessary for both establishment and maintenance of HCMV latency in HSC.

**Figure supplement 3**. KAP1 is necessary for establishment of HCMV latency in HSC.

**Figure supplement 4**. KAP1 is necessary for maintenance of HCMV latency in HSC.

effect on hematopoiesis (*Barde et al., 2013*), we could not exclude that its depletion induced HCMV lytic genes expression through indirect effects. We thus asked whether the co-repressor associates with the HCMV genome by performing KAP1-specific chromatin immunoprecipitation-deep sequencing (ChIP-seq) analyses on material isolated from human CD34+ HSC at 7 days post-TB40-E

infection. Using a stringent peak-calling algorithm and validating its results by ChIP-PCR, we identified 28 major KAP1-enriched regions on the HCMV genome (*Figure 2A,B* and *Supplementary file 1*). Supporting the functional significance of KAP1 recruitment to latent HCMV genomes, ChIP-PCR analyses also detected SETDB1, H3K9me3 and HP1α at these KAP1-enriched regions (*Figure 2C* and *Figure 2—figure supplement 1A*). Furthermore, in line with our former demonstration that heterochromatin formation can spread several tens of kilobases away from primary KAP1 docking sites (*Groner et al., 2010*), H3K9me3 and HP1α were found at significant distances from these major KAP1 peaks, and particularly covered the major immediate early promoter (MIEP), the viral origin of lytic replication (OriLyt) and the UL112 gene. In contrast, the repressive mark was not found at genes known to be expressed during latency such as UL138 or LUNA (*Reeves and Sinclair, 2013*) (*Figure 2D* and *Figure 2—figure supplement 1B*). Importantly, SETDB1 and HP1α recruitment as well as H3K9 trimethylation were lost when KAP1 was depleted by RNA interference prior to HSC infection, supporting a model whereby the corepressor recruits these heterochromatin-inducing factors (*Figure 2—figure supplement 1A–D*).

## An mTOR-mediated KAP1 phosphorylation switch between HCMV latency and lytic replication

We then examined the epigenetic status of HCMV genome when TB40-E-infected CD34+ cells were differentiated in mature dendritic cells (mDCs), a procedure previously demonstrated to result in HCMV activation. As expected, the TB40-E DNA was no longer associated with SETDB1 or HP1α and did not bear the H3K9me3 repressive mark in the CD34-derived mDCs (*Figure 3A* and *Figure 2—figure supplement 1A*). However, KAP1 was surprisingly still associated with the viral genome in these targets (*Figure 3B*). The same pattern was recorded in TB40-E infected MRC-5 cells, where the virus achieves a complete lytic cycle (*Figure 3—figure supplement 1A,B*). It suggested that HCMV reactivation and replication could occur in spite of corepressor binding. To probe this issue further, we infected CD34+ cord blood and MRC5 cells with the HCMV AD169 laboratory strain, which is incapable of inducing latency (*Goodrum et al., 2007*; *Saffert et al., 2010*). At day 7 post-infection, while the productively transcribed AD169 genome carried as expected neither SETDB1 nor H3K9Me3, it was still bound by KAP1 in both cell types as robustly as the latent TB40-E strain in HSC (*Figure 3—figure supplement 1C,D*). Therefore, it is not KAP1 recognition but rather the secondary recruitment of its SETDB1 effector and other KAP1-associated heterochromatin inducers such as HP1, which is responsible for HCMV latency.

Within the context of DNA damage, ATM phosphorylates KAP1 on serine 824, thereby reducing its ability to bind SETDB1 hence its repressor potential (*White et al., 2006*, *2012*; *Noon et al., 2010*). We thus asked whether this modification could alter the consequences of KAP1 recruitment to the HCMV genome. We could immunoprecipitate the TB40-E DNA with pS$^{824}$KAP1-specific antibodies in HCMV-reactivated mDC but not in latently infected HSC, with a pattern of genomic recruitment identical to that delineated with the global KAP1 antibody in both cell types (*Figure 3C,D*). Furthermore, pS$^{824}$KAP1 was similarly found associated with the replicating AD169 viral DNA in both MRC5 cells and CD34+ HSC (*Figure 3—figure supplement 1E,F*) and with the TB40-E-genome in productively infected MRC-5 fibroblasts (*Figure 3—figure supplement 1G*). Confirming these data, immunofluorescence detected the presence of pS$^{824}$KAP1 in 100% of TB40-E- or AD169-infected MRC-5 fibroblasts, but not in uninfected fibroblasts (*Figure 4A,B–D* and *Figure 4—figure supplement 1A*), nor in CD34+ HSC latently infected with the TB40-E strain, which could be identified by their loss of expression of the surface protein MRP1 as recently described (*Weekes et al., 2013*) (*Figure 4E*).

We then set to identify the kinase responsible for phosphorylating KAP1 in cells productively infected with HCMV. The ATM inhibitor KU55933 did not prevent this process, suggesting that this known KAP1 kinase did not play a primary role (*Figure 4—figure supplement 1B*). We thus turned to the mammalian target of rapamycin (mTOR), which was previously found activated in cells replicating HCMV (*Moorman and Shenk, 2010*; *Clippinger et al., 2011*; *Clippinger and Alwine, 2012*; *Poglitsch et al., 2012*). When TB40-E infected MRC-5 cells were treated with the mTOR inhibitors rapamycin or Torin1 (*Thoreen and Sabatini, 2009*; *Thoreen et al., 2009*), the phosphoKAP1-specific immunofluorescence signal was suppressed (*Figure 4C* and *Figure 4—figure supplement 1C*), whereas total levels of KAP1 were unaffected (*Figure 4—figure supplement 1D*). Torin1-preventable pS$^{824}$KAP1 accumulation was also documented in CD34+ HSC infected with the replicative AD169 strain (*Figure 4D* and *Figure 4—figure supplement 1E,F*). In this setting, the IE protein-specific

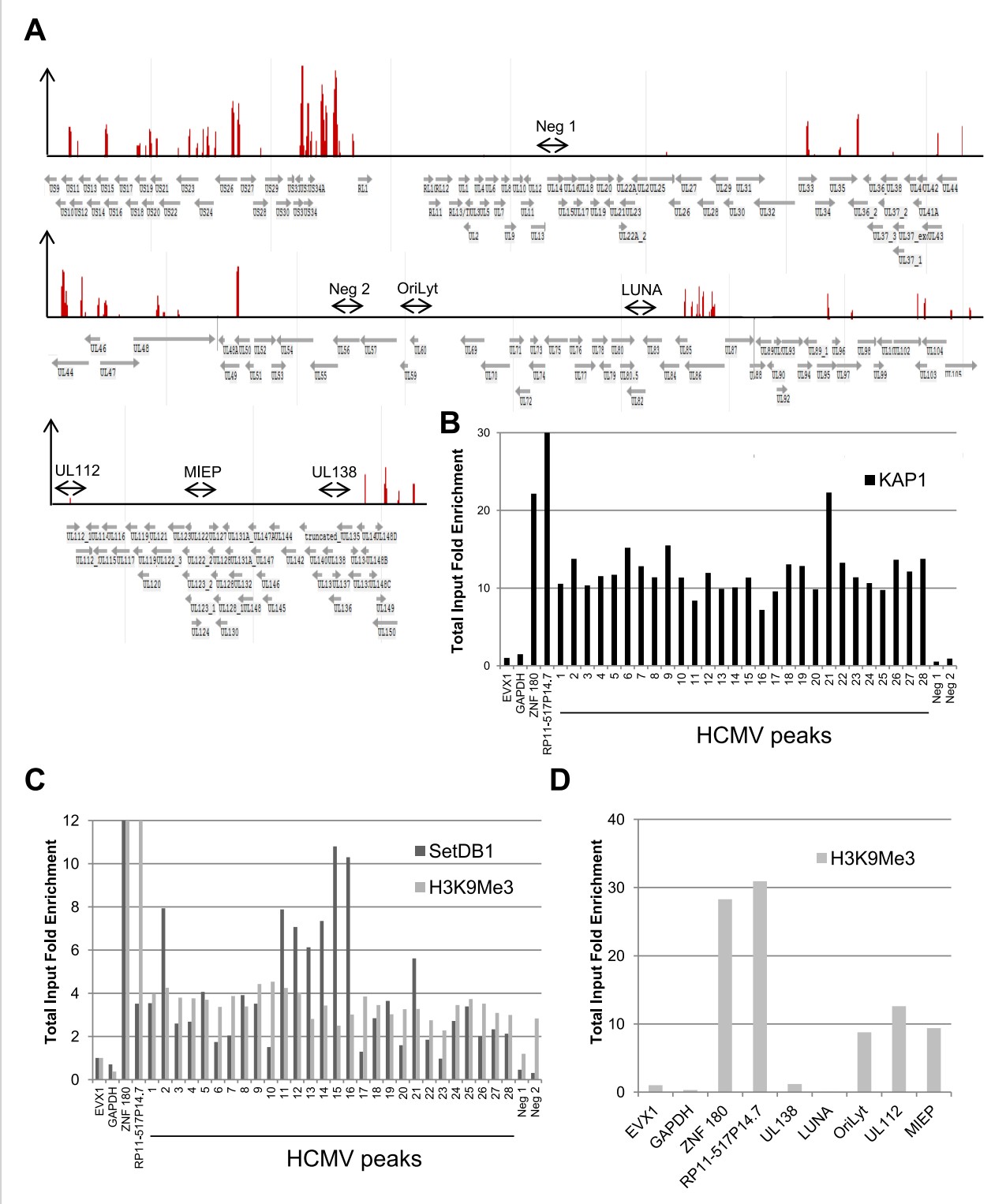

**Figure 2**. KAP1, SetDB1 and H3K9Me3 are enriched on the HCMV genome in latently infected HSC. (**A**) KAP1 binding sites on HCMV genome in TB40-E-infected CD34[+] cells, mapped by ChIP-Seq performed at day 7 post-infection. Results are presented as hit point upper 120 on the reference sequence GenBank EF999921.1, indicating all KAP1 peaks (1–28). (**B** and **C**) ChIP-PCR with anti-KAP1 (**B**), anti-SetDB1 and anti-H3K9Me3 (**C**) antibodies were performed on ChIP-Seq-mapped KAP1 peaks (1–28) plus two KAP1-negative HCMV regions (Neg 1 and 2), using EVX-1 and GAPDH as negative cellular gene controls, and ZNF180 and RP11-517P14.7 as positive controls. (**D**) H3K9me3-specific ChIP-PCR of two latency

*Figure 2. continued on next page*

*Figure 2. Continued*

genes (UL138, LUNA) and three regions active during lytic replication (OriLyt, UL112 and MIEP), with the same controls as in (**C**). Results are presented as total input fold enrichment (EVX-1 normalized).

The following figure supplement is available for figure 2:

**Figure supplement 1**. KAP1-dependent recruitment of HP1α and SETDB1 with H3K9 trimethylation of HCMV genome in latently infected HSC.

signal was reduced by the mTOR inhibitor, consistent with some repression of viral gene expression. Of note, some KAP1 phosphorylated on serine 473 was detected in MRC5 cells, but this was independent of HCMV infection and Torin1-resistant (*Figure 4—figure supplement 2A–C*). In contrast, recombinant mTOR could phosphorylate GST-KAP1 on $S^{824}$ in vitro (*Figure 4—figure supplement 2D*).

Cell fractionation analyses indicated that $pS^{824}$KAP1 accumulated in the nucleus of CMV-infected cells, which also displayed increased levels of phospho-S6 ribosomal protein, a marker of mTOR activation (*Figure 5A*). Indirect immunofluorescence and confocal microscopy further revealed mTOR-specific foci in the nucleus of TB40- or AD169-infected but not control MRC5 cells (*Figure 5B* and *Figure 5—figure supplement 1A*). Finally, ChIP with an mTOR-specific antibody demonstrated that the kinase associated with the CMV genome at the same places regions as $pS^{824}$KAP1 (*Figure 5C*). Consistent with a restriction of KAP1 phosphorylation and inactivation to CMV-associated molecules, the corepressor could still mediate the transcriptional silencing of an integrated TetO-PGK-GFP cassette via a Tet repressor-KRAB fusion protein (*Wiznerowicz and Trono, 2003*) in virus-infected cells (*Figure 5—figure supplement 1B*).

## Forcing HCMV out of latency by pharmacological manipulation of KAP1

These results suggested that HCMV silencing might be amenable to suppression by pharmacologically induced KAP1 phosphorylation. In the absence of strictly specific mTOR activator, we turned to ATM, a kinase previously demonstrated as capable of phosphorylating KAP1 on $S^{824}$ (*White et al., 2012*). When HSC infected 5 days earlier with TB40-E were incubated with the ATM activator chloroquine (*Bakkenist and Kastan, 2003*), intracellular levels of viral RNAs and DNA increased (*Figure 6A,B*), and infectious viral particles were released in the supernatant (*Figure 6C*). Supporting the specificity of the observed phenomenon, Torin-1 did not block chloroquine-induced HCMV RNA, DNA and viral particle production, whereas the ATM inhibitor Ku55933 prevented this process (*Figure 6*). These results obtained in a first set of HSC obtained from three donors were confirmed in another group of three donors, where it was further observed that levels of viral transcripts increased in drug-treated cells only transiently after a single dose of drug but steadily if three doses were administered over 5 days (*Figure 6—figure supplement 1A*), and that single doses of the ATM activator prevented the progressive drop in viral DNA that was observed in control cells, while repeated doses triggered an augmentation in viral DNA copies, indicative of genome replication (*Figure 6—figure supplement 1B*). Correlating these quantitative virological data, chloroquine treatment allowed the detection by immunofluorescence of $pS^{824}$KAP1-positive cells (*Figure 6—figure supplement 1C*). These were also systematically positive for IE antigens (*Figure 6—figure supplement 1C*), confirming that the virus harbored by these cells had exited latency. Finally, when the supernatant of TB40-E-infected HSC exposed to chloroquine was used to inoculate MRC5 fibroblasts, it induced the accumulation of viral DNA in these targets (*Figure 6—figure supplement 1D*), demonstrating that the production of replication-competent virus had been induced by ATM-activating treatment of the latently infected HSC. Noteworthy, $pS^{824}$KAP1 was detected only in HCMV IE-positive cells, which strongly suggests that KAP1 became phosphorylated in these cells through the combined action of ATM and some HCMV-encoded factor. Importantly, these effects were induced without drop in KAP1 levels or loss of surface expression of the CD34 stem cell marker (*Figure 6—figure supplement 2A,B*). Additional evidence strongly suggested that chloroquine stimulated HCMV via KAP1 phosphorylation. First, the drug did not further boost TB40-E in MRC-5 cells, where viral replication is KAP1-insensitive (*Figure 6—figure supplement 2C,D*). Second, it did not further stimulate HCMV gene expression and virion release in CD34+ cells depleted for KAP1, whether or not these were complemented

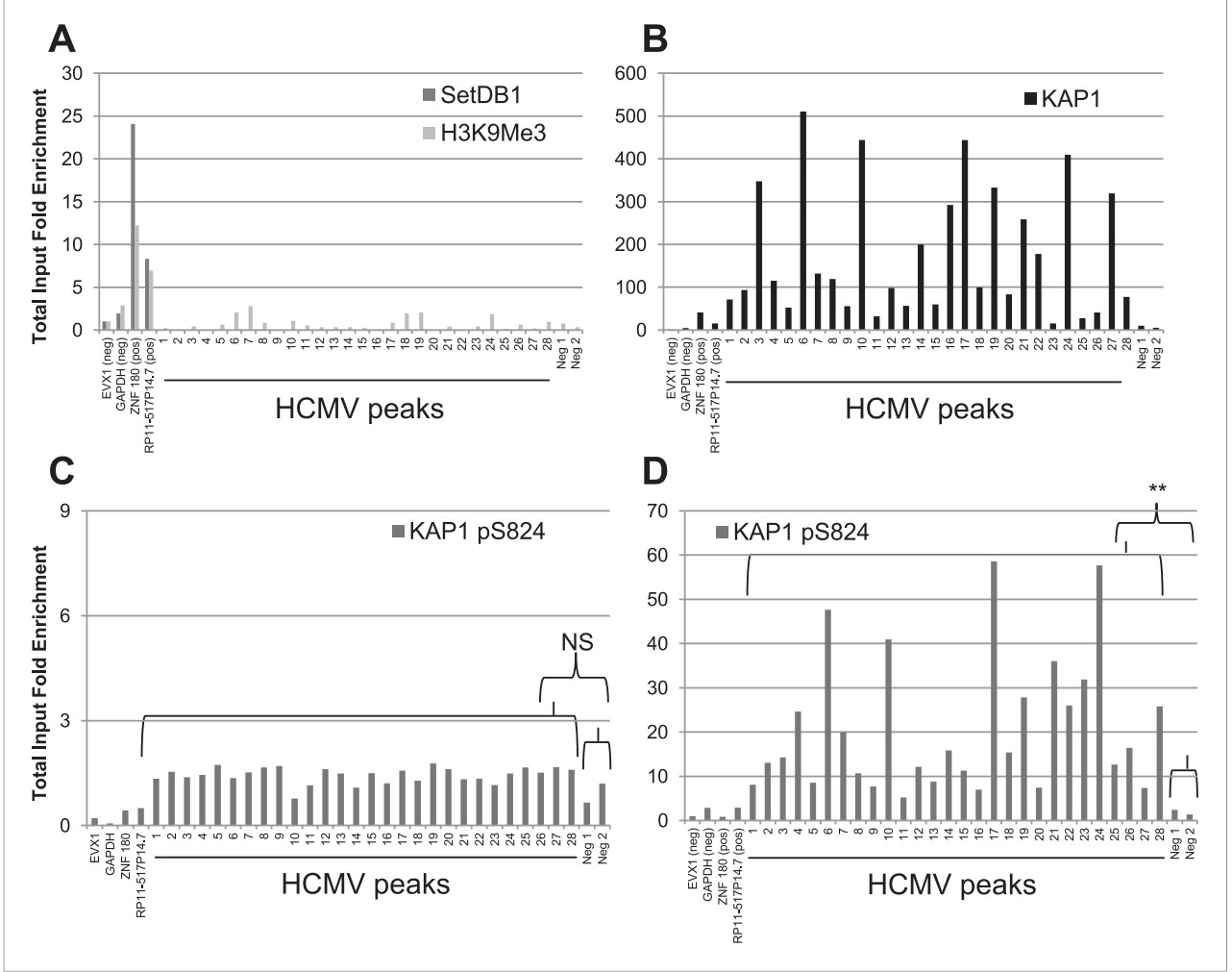

**Figure 3**. A KAP1 phosphorylation switch governs HCMV progression from latency to lytic replication. ChIP-PCR of indicated HCMV genomic regions was performed with anti-SetDB1 and anti-H3K9Me3 (**A**), anti-KAP1 (**B**) or anti- S824 phosphoKAP1 antibodies (**C** and **D**) on material extracted from TB40-E-infected cells CD34+ cells (**C**) or mature dendritic cells derived therefrom (**A**, **B**, **D**), using same controls as in *Figure 2*. Results are presented as total input fold enrichment after EVX-1 normalization.

The following figure supplement is available for figure 3:

**Figure supplement 1**. A KAP1 phosphorylation switch distinguishes HCMV latency and lytic replication.

with a phosphorylation resistant S$^{824}$A KAP1 mutant (*Figure 6—figure supplement 3A,B* and *Figure 6—figure supplement 4AB*). Third, and by contrast, chloroquine did reactivate HCMV when these knockdown cells were complemented with wild-type KAP1 (*Figure 6—figure supplement 4A,B*).

We also asked whether the pharmacological induction of KAP1 phosphorylation could trigger HCMV production from circulating monocytes. Monocytes were purified from the peripheral blood of seropositive individuals and exposed to three doses (at days 0, 3 and 5) of chloroquine, alone or in combination with Torin-1 or KU55933 (*Figure 7*). ATM activation did not trigger the differentiation of the monocytes into dendritic cells or macrophages, as verified by measuring the cell surface expression of CD1a, CD14, CD80 and CD86 (*Figure 7—figure supplement 1*). However, it induced the intracellular accumulation of HCMV DNA and the release of infectious viral particles from these monocyte populations. Furthermore, Ku55933 but not Torin-1 prevented the stimulating effect of chloroquine, as noted in HSC.

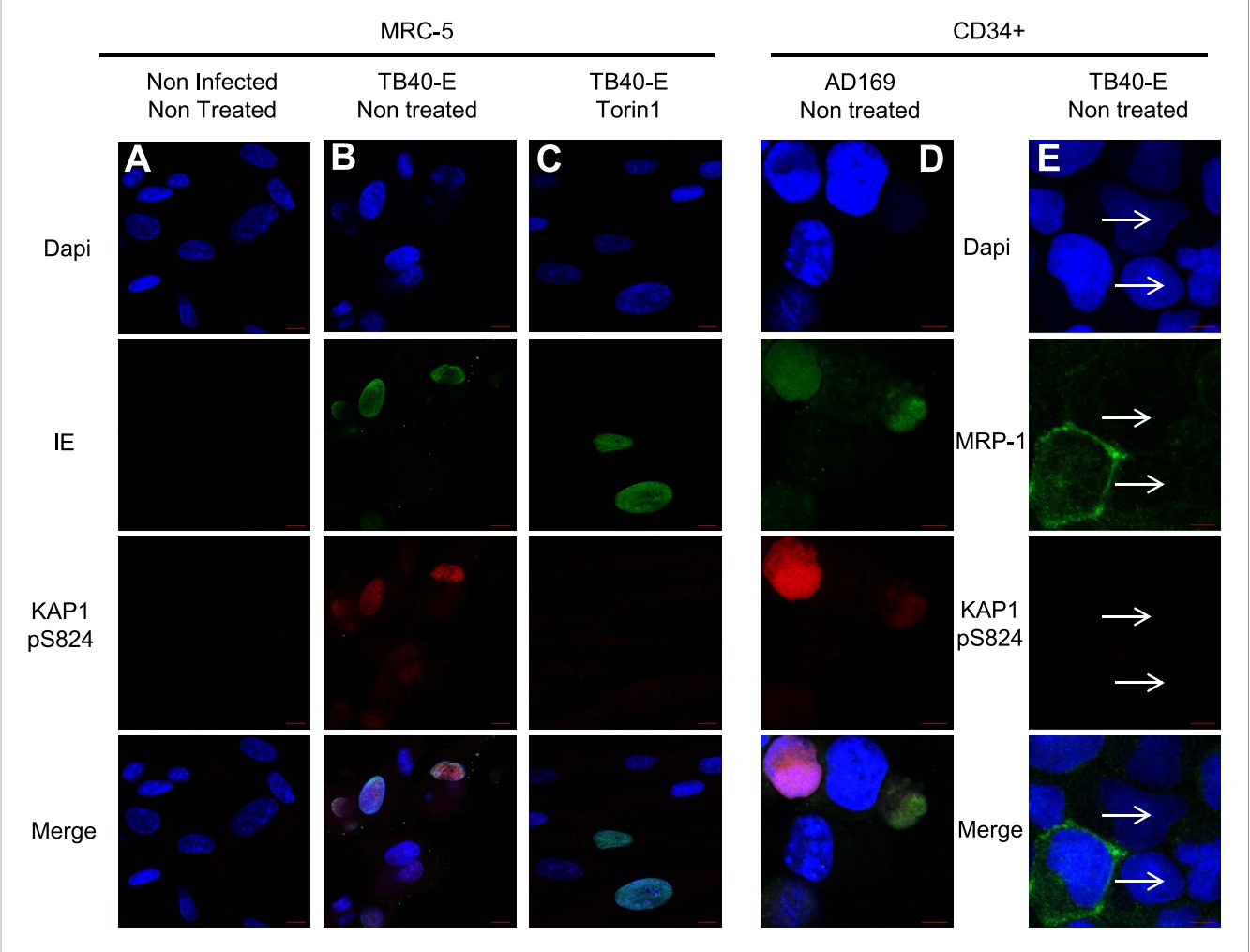

**Figure 4**. mTOR inhibition prevents HCMV-induced KAP1 phosphorylation. MRC-5 (**A**, **B**, **C**) and CD34[+] (**D** and **E**) cells, infected (AD169 or TB40-E) or not (Non Infected) with HCMV, and treated or not (Non Treated) with mTOR inhibitor (Torin1) were examined by immunofluorescence with anti-IE and anti-MRP1 antibodies (Alex-488, green), and anti-S824 phosphoKAP1 antibodies (Alexa 568, red). DNA was stained with Dapi (blue). White arrows represent latently infected cells (MRP-1 negative cells). Scale Bar in white represents 10 μm for MRC-5 and 5 μm for CD34[+]. All pictures are representative of slide overview from three independent experiments.

The following figure supplements are available for figure 4:

**Figure supplement 1**. mTOR and HCMV-associated KAP1 phosphorylation.

**Figure supplement 2**. mTOR and HCMV-associated KAP1 phosphorylation.

These data demonstrate that inducing KAP1 phosphorylation results in releasing HCMV from its latent state by relieving repressive marks on the viral chromatin. Nevertheless, levels of viral gene expression achieved in this setting remain lower than observed when HCMV-harboring precursors are differentiated into macrophages or mature dendritic cells, where NF-κB has been demonstrated to drive viral transcription. We thus exposed latently infected CD34[+] cells to chloroquine and TNF- α (5 ng/ml), either alone or in combination (*Figure 8*). With TNF alone, HCMV transcription was not induced, indicating that the virus remained latent. With chloroquine alone, significant induction of cell-associated viral transcripts (approx. 100-fold) and DNA (approx. fivefold) was measured, and viral particles were released in the supernatant (10,000 PFU/ml in the experiment depicted in *Figure 8*). With the further addition of TNF, levels of CMV-specific RNAs increased by another 10-fold, the amounts of cell-associated viral DNA doubled, and virion production was boosted by a factor 10.

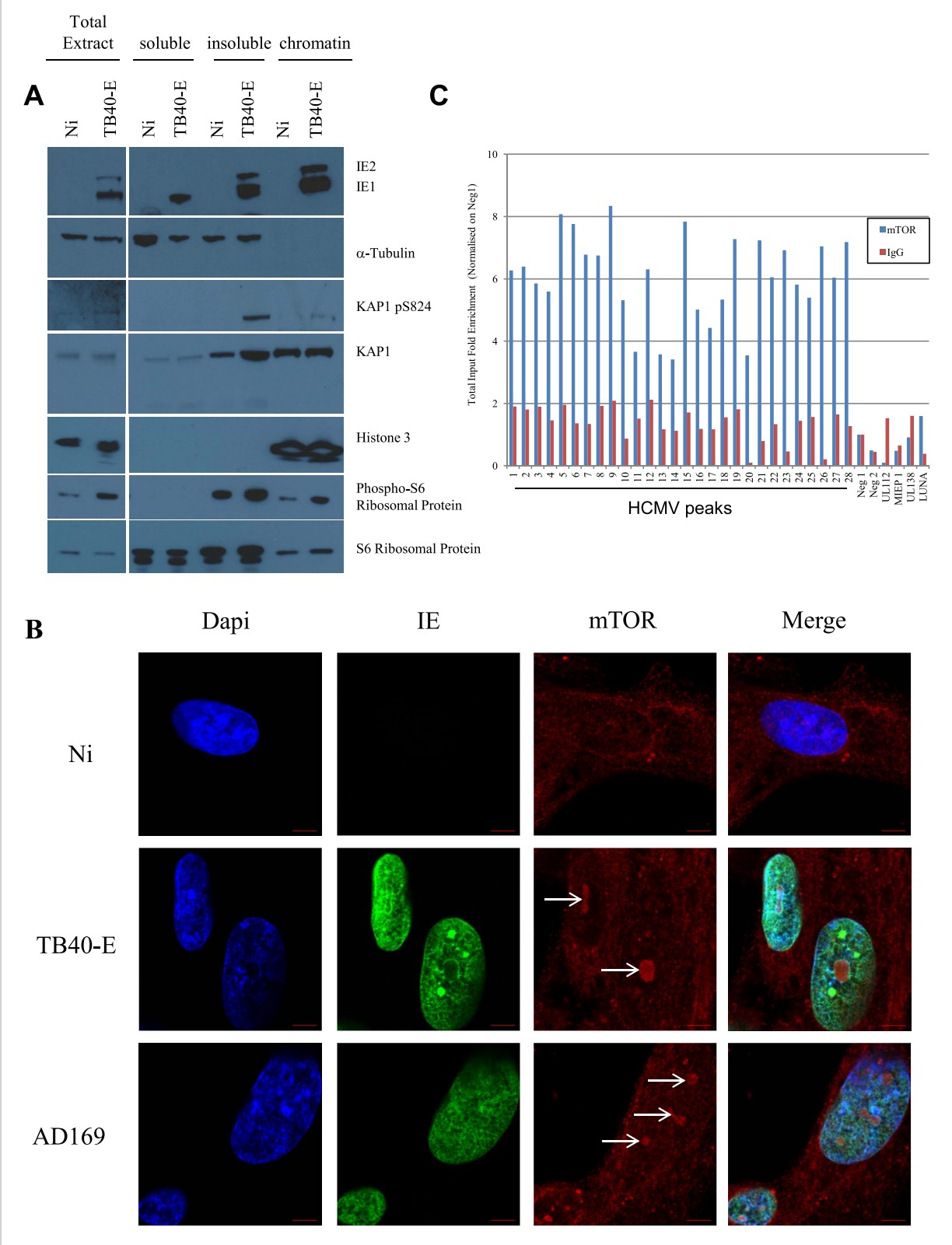

**Figure 5**. mTOR associates with the HCMV genome during lytic replication. (**A**) MRC-5 cells were infected 3 days (TB40-E) or not (NT) with HCMV and harvested for cell fractionation. IE1/2, α-Tubulin, KAP1, KAP1 pS824, Histone 3 and phosphorylated or not S6 ribosomal protein expression/localization in Total extract, cytosolic (soluble) nuclear (insoluble) and chromatin (chromatin) parts were analysed by Western-Blot. (**B**) Immunofluorescence by confocal microscopy was performed on MRC-5, HCMV infected (TB40-E, AD169) or not (Non Infected). Staining was performed with anti-IE (Alex-488, green), and

*Figure 5. continued on next page*

Figure 5. Continued

anti-mTOR antibodies (Alexa 568, red). DNA was stained with Dapi (blue). Scale Bar in white represents 5 μm. (**C**) ChIP-PCR for indicated HCMV genomic regions were performed with anti-mTOR or control-IgG on material extracted from TB40-E-infected MRC-5. Results are presented as total input fold enrichment after HCMV Neg1 region normalization.

The following figure supplement is available for figure 5:

**Figure supplement 1**. mTOR nuclear translocation and KAP1 activity in HCMV replicating cells.

Importantly, this dual treatment did not trigger the differentiation of the HSC (*Figure 8—figure supplement 1*). Taken together, these results suggest a model whereby chloroquine treatment renders the HCMV chromatin permissive for transcription through KAP1 $S^{824}$ phosphorylation, which then allows TNF to induce full viral gene expression via NF-kB activation.

## Discussion

This work sheds light on the mechanisms of HCMV persistence by revealing that, in latently infected hematopoietic stem cells, the master co-repressor KAP1 recruits HP1α and SETDB1 to the viral genome, triggering H3K9 methylation and heterochromatin formation. Our results correlate with the observation that, in latently infected CD34[+] HSC or circulating monocytes isolated from seropositive individuals, chromatin at the HCMV MIEP bears H3K9me3 and HP1 (*Sinclair, 2010*). Our finding that KAP1 partakes in both the establishment and the maintenance of HCMV latency also corroborates the previously noted absence of more permanent silencing marks, such as DNA methylation, on latent HCMV genomes (*Hummel et al., 2007*). We provide another important clue to the understanding of HCMV biology by revealing that the virus exits latency when KAP1 becomes phosphorylated on serine 824, which blocks its ability to recruit SETDB1 hence abrogates its co-repressor potential. We further identify mTOR as responsible for this KAP1 phosphorylation switch, and capitalizing on these findings we finally demonstrate that HCMV can be forced out of latency by pharmacological activation of ATM, another KAP1 kinase, and that this effect can be potentiated by adding TNF, an inducer of NF-kB. Whether KAP1 phosphorylation relieves HCMV from latency only by dislodging SETDB1 and HP1 from the viral chromatin or through more active mechanisms remains to be determined. Interestingly, a small molecule inhibitor of LSD1/KDM1A was recently found to decrease HCMV immediate early gene transcription (*Liang et al., 2013*). It could be that pS$^{824}$KAP1 plays a role in attracting this H3K9 demethylase to the viral genome.

There are remarkable analogies but also interesting differences between our results and recent data implicating KAP1 in the control of KSHV latency (*Chang et al., 2009*; *Cai et al., 2013*). In cells harboring latent KSHV genomes, KAP1, HP1 and H3K9me3 were enriched at more than two thirds of viral promoters, and KAP1 knockdown reduced these repressive marks and stimulated virus production. Furthermore, upon induction of a lytic cycle by overexpression of the Rta viral transactivator, KAP1 dissociated from the viral genome, resulting in loss of HP1 and H3K9me3. This correlated with phosphorylation of KAP1 on $S^{824}$, apparently mediated by the viral kinase vPK, with reciprocal decrease in the level of KAP1 sumoylation (*Chang et al., 2009*). Here, while we find that a KAP1 phosphorylation switch can also force HCMV out of latency, we demonstrate that, with this virus, the phosphorylated form of the master repressor remains associated with actively transcribed viral genomes. Furthermore, we implicate the cellular kinase mTOR in KAP1 phosphorylation. It was previously noted that both the mTORC1 (mTOR complex 1) and mTORC2 (mTOR complex 2) arms of the mTOR pathway are activated during lytic HCMV replication (*Kudchodkar et al., 2004*, *2006*, *2007*). This results in phosphorylating notably the eukaryotic initiation factor 4E (eIF4E)-binding protein (4E-BP1) and the p70S6 kinase (S6K), which enhances viral protein translation and more generally minimizes cellular stress responses (*Clippinger et al., 2011*; *Poglitsch et al., 2012*). Here, we identify KAP1 as another substrate of this cascade, by observing that (i) the co-repressor is phosphorylated on serine 824 in cells productively infected with HCMV; (ii) this process is inhibited by the mTOR inhibitors rapamycin and Torin1; and (iii) mTOR targets the viral chromatin at KAP1-bearing genomic sites during a lytic cycle. Conversely, we could force HCMV out of transcriptional dormancy by inducing the activation of ATM, another kinase targeting KAP1 on serine 824, in latently infected HSC and circulating monocytes. Nevertheless, our data also point to the role of viral factors in targeting this process. First, $S^{824}$ phosphoKAP1 was

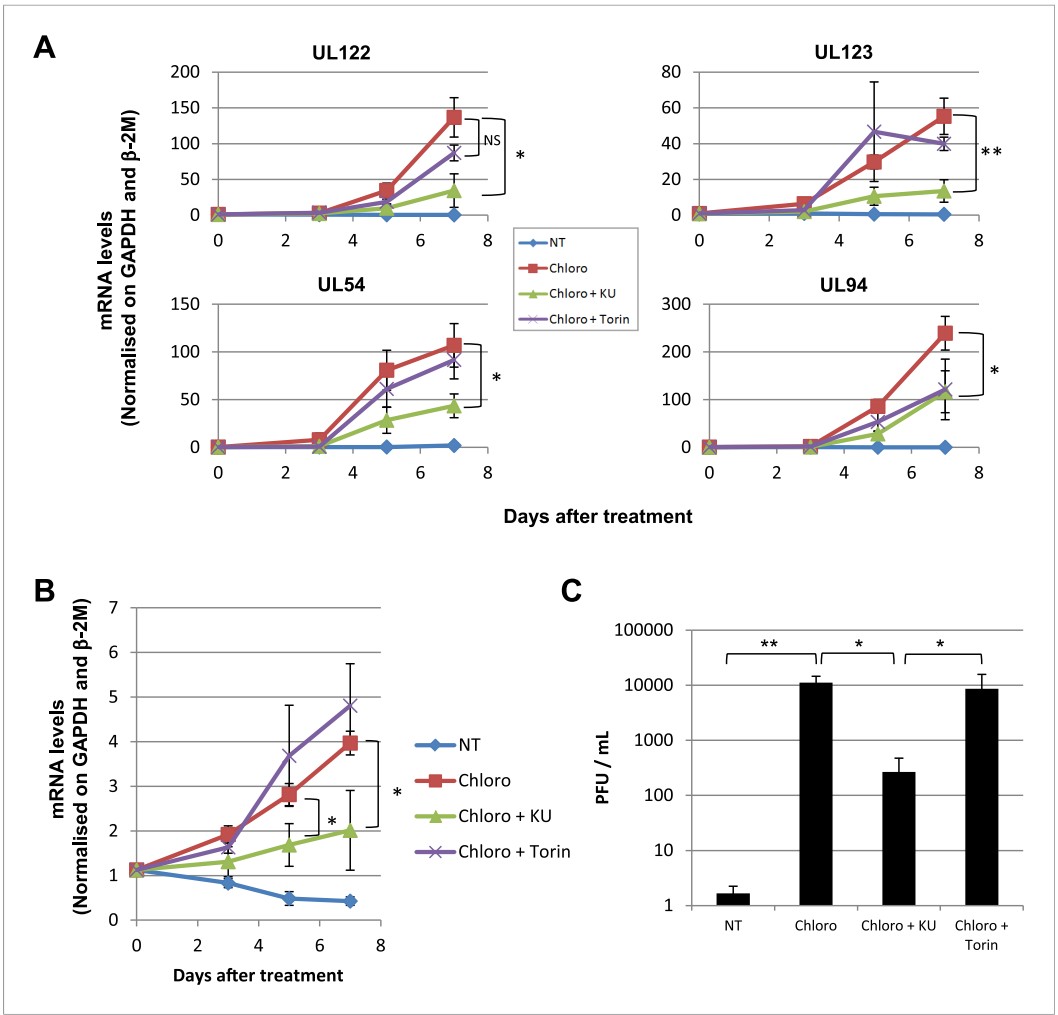

**Figure 6**. Forcing HCMV out of latency in CD34[+] cells by KAP1 pharmacological manipulation. After 5 days of infection with TB40-E, CD34[+] cells were treated three times (Day 0, 3 and 5) or not (NT) with Chloroquine (Chloro) alone or in combination with the mTor inhibitor Torin-1 (Torin) or the ATM inhibitor KU59933 (KU). (**A**) Indicated HCMV transcripts were quantified by RT-qPCR, using GAPDH and β-2 microglobulin for normalization. (**B**) HCMV DNA associated with the TB40-E-infected HSC was quantified by qPCR, normalizing with the GAPDH and albumin genes. Data are presented as average of three different experiments performed with cells from independent donors. (**C**) Supernatant from the TB40-E-infected CD34[+] cells, harvested after 7 days of treatment and the viral production was quantified by classical plaque assay on MRC-5 fibroblasts. Results are presented as PFU/ml. Histogram represents an average of three different experiments performed with HSC from three independent donors (n = 3, *p < 0.05, **p < 0.01, error bars as s.d.). See *Figure 6—figure supplements 5, 6* for all individual experiments.

The following figure supplements are available for figure 6:

**Figure supplement 1**. HCMV can be forced out of latency in CD34[+] cells by pharmacological manipulation.

**Figure supplement 2**. HCMV can be forced out of latency in CD34[+] cells by pharmacological manipulation.

**Figure supplement 3**. HCMV can be forced out of latency in CD34[+] cells by pharmacological manipulation.

**Figure supplement 4**. HCMV can be forced out of latency in CD34[+] cells by pharmacological manipulation.

**Figure supplement 5**. HCMV can be forced out of latency in CD34[+] cells by pharmacological manipulation.

**Figure supplement 6**. HCMV can be forced out of latency in CD34[+] cells by pharmacological manipulation.

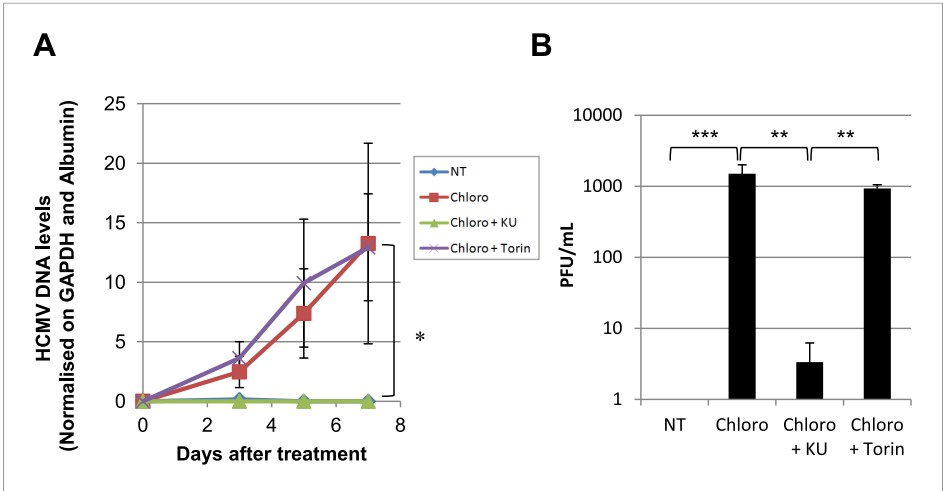

**Figure 7**. Inducing KAP1 phosphorylation releases HCMV from latency in monocytes. Monocytes from HCMV seropositive donors were purified and treated three times (Day 0, 3 and 5) or not (NT) with Chloroquine (Chloro) alone in combination with Torin-1 (Torin) or KU59933. (**A**) HCMV DNA associated with the monocytes was quantified by qPCR, normalizing with the GAPDH and albumin genes. Data are presented as average of three different experiments performed with cells from independent donors. (**B**) Supernatant from the monocytes, harvested after 7 days of treatment and the viral production was quantified by classical plaque assay on MRC-5 fibroblasts. Results are presented as PFU/ml. Histogram represents an average of three different experiments performed with HSC from three independent donors (n = 3, *p < 0.05, **p < 0.01, error bars as s.d.). See *Figure 7—figure supplement 2* for all individual experiments.

The following figure supplements are available for figure 7:

**Figure supplement 1**. Monocytes do not differentiate during pharmacological reactivation of HCMV.

**Figure supplement 2**. Monocytes do not differentiate during pharmacological reactivation of HCMV.

detected only in HCMV IE antigen-positive cells. Second, KAP1 conserved its ability to repress a chromosomal target in cells where its HCMV silencing activity was abrogated by phosphorylation. This strongly suggests that mTOR is targeted to the HCMV genome, where the viral DNA-associated KAP1 molecules are then exposed to its action, a model supported by the results of our mTOR-specific chromatin immunoprecipitation. Such spatial compartmentalization would not be unprecedented, as site-specific KAP1 phosphorylation is also observed in case of DNA damage (*White et al., 2006*), and as mTOR has itself been observed to be directed to sites of HCMV assembly (*Clippinger and Alwine, 2012*). More generally, it will be interesting to ask whether the KRAB/KAP1 pathway contributes to the latency of other herpes viruses, even though in the case of EBV the only evidence available so far is that it instead promotes viral DNA replication (*Liao et al., 2005*).

Latency is a dominant, cell type- and differentiation stage-specific mechanism (*Saffert et al., 2010*; *Goodrum et al., 2012*), as it is limited to HSC and cells of the myeloid lineage such as monocytes, and to HCMV clinical isolates (e.g., TB40-E) that contain genomic regions commonly deleted in latency-incompetent laboratory strains (e.g., AD169) (*Goodrum et al., 2007*; *Reeves and Sinclair, 2010*; *Petrucelli et al., 2012*). Our findings thus suggest that, in HSC, a latency-specific gene product counters HCMV-induced mTOR-mediated KAP1 phosphorylation, for instance by either blocking the kinase or by acting as or inducing a KAP1-specific phosphatase. This warrants investigations aimed at elucidating this aspect, and at determining whether, when and, if so, downstream of what pathway mTOR becomes activated and KAP1 phosphorylated during the differentiation of HSC towards macrophages or activated DCs, the cells in which HCMV transcription becomes fully productive.

KAP1 is not a DNA-binding protein, and is classically tethered to given genomic loci, for instance endogenous retroelements, by sequence-specific KRAB-ZFPs (*Friedman et al., 1996*; *Wolf and Goff, 2009*; *Rowe and Trono, 2011*; *Castro-Diaz et al., 2014*). By analogy, it is tempting to hypothesize

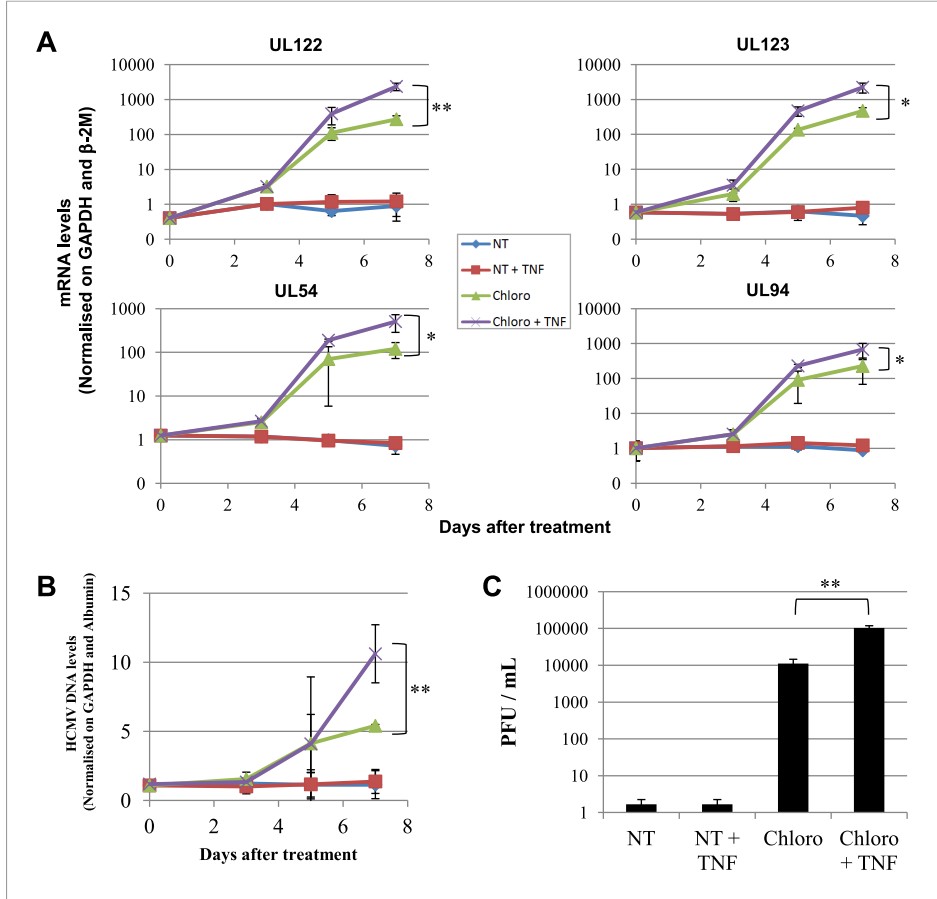

**Figure 8**. Combining KAP1 phosphorylation and NF-κB induction increases HCMV activation from latently infected HSC. After 5 days of infection with TB40-E, CD34[+] cells were treated three times (Day 0, 3 and 5) or not (NT) with chloroquine (Chloro) in combination or not with 5 ng/ml of recombinant TNF-α (TNF). (**A**) Indicated HCMV transcripts were quantified by RT-qPCR, using GAPDH and β-2 microglobulin for normalization. (**B**) HCMV DNA associated with the TB40-E-infected HSC was quantified by qPCR, normalizing with the GAPDH and albumin genes. Data are presented as average of three different experiments performed with cells from independent donors. (**C**) Supernatant from the TB40-E-infected CD34[+] cells, harvested after 7 days of treatment and the viral production was quantified by classical plaque assay on MRC-5 fibroblasts. Results are presented as PFU/ml. Histogram represents an average of three different experiments performed with HSC from three independent donors (n = 3, *p < 0.05, **p < 0.01, error bars as s.d.). See **Figure 8—figure supplements 2, 3** for all individual experiments.

The following figure supplements are available for figure 8:

**Figure supplement 1**. Pharmacological activation of HCMV does not trigger CD34[+] cells differentiation.

**Figure supplement 2**. Pharmacological activation of HCMV does not trigger CD34[+] cells differentiation.

**Figure supplement 3**. Pharmacological activation of HCMV does not trigger CD34[+] cells differentiation.

that members of the KRAB-ZFP family are also involved in recognizing the HCMV DNA. In support of this model, KAP1 recruitment to the HCMV genome was lost when the KRAB-binding RBCC domain was deleted (not illustrated). Since we detected close to thirty major KAP1 peaks on the HCMV DNA and failed to identify a common motif amongst all these sequences, more than one DNA-binding protein, whether or not KRAB-ZFP, is likely involved in docking the corepressor at these sites. In the case of KSHV, the latency protein LANA is responsible for tethering sumoylated KAP1 and the KAP1-bound Sin3A repressor to the Rta promoter, through its ability to interact with SUMO-2-bearing proteins. Upon hypoxia, KAP1 is de-sumoylated and together with Sin3A is released from the KSHV genome, leading to

the activation of the LANA-associated HIF-1α, the inducible subunit of the hererodimeric HIF-1 (hypoxia-induced factor 1), thus activating viral gene expression (*Cai et al., 2013*). By analogy, it could be that a virally encoded product contributes to targeting KAP1 to the HCMV genome.

Our data have important clinical implications. First, it was recently noted that transplant recipients placed on immunosuppressive regimens that included rapamycin were less likely to suffer serious CMV-related complications than patients for whom this drug was omitted (*Andrassy et al., 2012*; *Sabé et al., 2012*). The present work indicates that this beneficial effect might stem at least in part from a prevention of viral reactivation. This warrants further studies to assess the positive impact of this and other mTOR inhibitors on the control of HCMV infection, whether after transplantation or in other settings. Of note, a recent study demonstrated that rapamycin did not prevent HCMV reactivation from latently infected dendritic cells in vitro (*Glover et al., 2014*). However, it could be that KAP1 phosphorylation was already established in this system, and that maturation of the DCs only resulted in activating transcription factors that are strong stimulators of HCMV gene expression, such as NF-κB.

Our discovery that HCMV can be forced out of latency by pharmacological manipulation suggests new avenues to eradicate infection by combining ATM and NF-κB activation with immune- or drug-based approaches aimed at killing HCMV-infected cells. The systemic administration of ATM and NF-κB activators may cause important side effects, and HCMV eradication is anyway probably unnecessary in immuno-competent individuals. However, the ex vivo treatment of HCMV-positive transplants, for instance bone marrow or HSC purified therefrom, could be readily envisioned to purge latently infected cells prior to engraftment, notably in the high-risk setting of HCMV-negative recipients.

## Materials and methods

### Cell culture and treatments

CD34$^+$ cells from HCMV negative human cord blood were obtained from the Lausanne University Hospital (Centre Hospitalier Universitaire Vaudois, CHUV, Switzerland) delivery room with proper informed consent, layered on Ficoll gradient, purified with the CD34$^+$ cell separation kit (Miltenyi Biotec, Germany) and freshly used for experiments without freezing to avoid differentiation. CD34$^+$ cells were stimulated for one day in X-Vivo 15 medium (Lonza, Switzerland) supplied with cytokines (100 ng/ml of Flt-3 ligand, 100 ng/ml of SCF, 20 ng/ml of TPO, and 20 ng/ml of IL-6; Peprotech, Rocky Hill, NJ) and 5% penicillin/streptomycin, maintained at a density of $5.10^5$ cells/ml and transduced the day after stimulation with lentiviral vector at a MOI of 100 as previously described (*Barde et al., 2013*), washed 24 hr later and cultured in X-Vivo 15 medium, 50 ng/ml of Flt-3 ligand, 25 ng/ml of SCF and 20 ng/ml of TPO. MRC5 fibroblasts serum and 5% penicillin/streptomycin. MRC-5 fibroblasts were transduced with a lentivector at a MOI of 10, 3 days before HCMV infection. Monocytes were purified from HCMV seropositive donors buffy coat, obtained at the EFS (Etablissement Français du Sang, France), with CD14$^+$ magnetic selection kit (Miltenyi Biotec). After 7 days of HCMV reactivation treatment, non differentiation of cells was controlled by different marker expression by FACS (CD1a, CD14, CD80, CD86) (Invitrogen, Carlsbad, CA). Cells were cultured in RPMI medium (invitrogen) supplemented with 10% FCS and Peni/strep. The mTOR (Torin1 and rapamycin, Selleckchem, Houston, TX) and ATM (KU55933, LuBioscience, Switzerland) inhibitors were used at 0.1 and 10 μM, respectively, added 30 min before infection. The ATM activator (chloroquine, Sigma–Aldrich, St Louis, MO) was used at 20 μM. The recombinant TNF-α (Peprotech) is used at 5 ng/ml.

### Mature DC generation

After 10 days of TB40-E infection, CD34$^+$ HSC were cultured for 7 days in X-VIVO-15 medium supplemented with GM-CSF (100 ng/ml), TGF-β (0.5 ng/ml), Flt-3 ligand (100 ng/ml), SCF (20 ng/ml) and TNF-α (50 U/ml) (all from peprotech). This treatment leads to generation of immature dendritic cells that will be pushed to maturation with lipopolysaccharide (LPS, 50 ng/ml) treatment during 3 days, as described in (*Reeves et al., 2005*).

### Ethics statement

Cord blood was obtained after approval of the project by the local ethics committee, commission cantonale d'ethique de la recherche sur l'etre humain, as protocol 146/10. When necessary, samples were obtained following informed consent of the participants.

## Lentiviral vectors

pLKO vectors were purchased from Sigma–Aldrich and the puromycin was replaced by eGFP (pLKO-shKAP1, pLKO-empty, pLKO-scramble). QuickChange II XL Site directed Mutagenesis Kit (Agilent Technologies Santa Clara, CA) was used to mutate KAP1 Serine 824 into Glutamic Acid in a pFUT plasmid containing a human *Kap1* allele modified by nucleotide substitutions to resist to our shRNA. Production and titration of lentiviral vectors was performed as previously described (*Barde et al., 2011*).

## Viruses

The AD169 (ATCC, Manassas, VA) and TB40-E (a gift from G Herbein, Besançon, France) HCMV strains were used in this study. Virus stocks were generated in MRC5 or HUVEC cells, collecting particles when cytopathic effects were >90%. Supernatants were clarified of cell debris by centrifugation at 1,500×$g$ for 10 min, ultracentrifuged at 100,000×$g$ for 30 min at 4°C and stored at −80°C until use. Virus titers were determined by plaque assay on MRC5 cells using standard methods (MEM2X diluted twofold with solution of 1.6% agarose). MRC-5 were infected at a MOI of 1, while CD34$^+$ infection were performed at a MOI of 5, 3 days after or 7 days before transduction with lentivectors.

## Chromatin immunoprecipitation (ChIP)

Chromatin from 10$^7$ CD34$^+$ was prepared and immunoprecipitated as in (*Barde et al., 2013*), with KAP1- (Abcam, UK), H3K9me3- (Diagenode, Belgium), SETDB1- (Abcam) mTOR (Abcam) or S$^{824}$ phosphoKAP1- (Abcam) specific antibodies. TB40-E infected CD34$^+$ KAP1-ChIPed DNA was sent to sequencing. The 50 bp reads for KAP1 IP and Total Input were mapped to the HCMV genome (TB40-BAC4 GenBank accession EF999921.1) using the bowtie short read aligner version 0.12.7 (*Langmead et al., 2009*) and allowing up to three mismatches over the whole length of the read. The signal of the total input was then subtracted from the KAP1 IP signal using a custom made PERL program (see *Source code 1*) and regions with similar signal were merged together. The resulting regions were ranked according to their signal. The signal follows a normal distribution (tested with Shapiro–Wilk normality test) thus we considered as enriched regions the extreme right part of the distribution. The data discussed in this publication have been deposited in NCBI's Gene Expression Omnibus and are accessible through GEO Series accession number GSE53271 (http://www.ncbi.nlm.nih.gov/geo/query/acc.cgi?acc=GSE53271). SYBR green qPCR was performed to quantify enrichment at specific loci. Total Input Fold enrichment was quantified by the classical following calculation: $100 \times 2^{-(CT \text{ of ChIPed DNA} - CT \text{ of Total Input})}$. Cellular promoters regions were used as negative (EVX-1; GAPDH) or positive (ZNF-180; RP11-517P14.7) controls for PCR enrichment. 28 highly enriched peaks were found for the HCMV TB40-E strain, and two negative regions were designed for qPCR controls.

## ImmunoFluorescence

MRC-5 were cultured directly on coverslips. Around $2 \times 10^5$ CD34$^+$ were attached to slides by cytospinning with Shandon EZ Single Cytofunnel (Thermo Fisher Scientific, Waltham, MA). Cells were then fixed and permeabilised with Methanol during 5 min at −20°C, and saturated with PBS 5% FCS during 2 hr at room temperature (RT). Primary staining with IE-, anti-S$^{824}$ phopshoKAP1-, KAP1- mTOR (all from Abcam) anti-KAP1 S$^{473}$ phosphoKAP1 (BioLegend, San Diego, CA) specific antibodies was performed in PBS 5% FCS at 4°C overnight or 3 hr at RT, before addition of secondary antibodies (Alexa-488 or -565) in PBS 5% FCS during 1 hr at RT. Dapi bath was performed during 10 min at RT, and slides were mounted with Fluoromount-G (Southern Biotech, Birmingham, AL).

## RT-PCR

Cells were sorted for GFP (control of transduction) and CD34 expression. RNA was extracted by Trizol and reverse transcribed with Superscript II (both from Invitrogen) according to the manufacturer's instructions. All qPCR were performed with SYBR green mix (Roche, Switzerland). Primers used in this work were designed by Primer Express software (Applied Biosystems) and their sequences are available on *Supplementary file 2*.

## Cell fractionation

Cells were washed two times with cold PBS and centrifuged. Pellets were resuspended in 500 µl of cytop Buffer (Triton 0.25%, Tris HCl 10 mM, EDTA 5 mM, EGTA 0.5 mM and proteases

inhibitor cocktail) and incubated 3–5 min on ice. After centrifugation, supernatant was kept as soluble part (cytoplasm) and pellet were lysed with TNEN 250:0.1 buffer (NaCl 250 mM, Tris HCl 50 mM, EDTA 5 mM, NP40 0.1% and proteases inhibitor cocktail) 20–30 min on ice. After centrifugation, supernatants were kept as insoluble part (nucleus) and pellets were lysed in Urea 8 M, 10 min at 95°C to recover chromatin part. Total amount of protein was quantified by BCA assay.

## Western blot analyses

Cells lysates were subjected to SDS/PAGE on 4–12% polyacrylamide gels (Invitrogen). iBlots on Nitrocellulose membrane (Invitrogen) were treated with KAP1- Histone 3- α-Tubulin- (Abcam) mTOR-S6 ribosomal protein- phospho S6 ribosomal protein (Cell Signaling, Beverly, MA) and βactin-(Calbiochem, San Diego, CA) specific antibodies, followed by polyclonal rabbit HRP-conjugated antibody, and protein bands were detected by using an ECL-plus kit (Thermo-Scientific). Anti-KAP1pS824 antibody was produced in rabbits using a KLH-MBS coupled peptide with the following sequence: AcNH-CAG LSS QEL pSGG P-CONH2 from Eurogentec.

## In vitro kinase assays

GST-KAP1 and 1 U of mTOR (475987 Millipore, Billerica, MA) were incubated in phosphorylation buffer (25 mM Hepes pH 6.8, 10 mM $MgCl_2$, 0.5 mM DTT, 200 µM ATT, 1× Complete protease inhibitor cocktail from Roche, 1% BSA) in a total volume of 15 µl for 30 min with shaking at room temperature. mTOR activity is inhibited with 100 nM of Torin1.

## Acknowledgements

This work was supported by grants from the Swiss National Science Foundation and the European Research Council. We thank the staff and patients of the CHUV delivery room for providing us with umbilical cord blood, the personnel of our core facilities for technical expertise, Julien Marquis for his help with the initial ChIP-Seq studies and Els Verhoyen (UMR-S 758-VIR, Lyon, France), Georges Herbein (Laboratoire de Virologie, Hôpital St Jacques, Besançon, France) and Christian Davrinche (UMR1043, centre de physiopathologie de Toulouse Purpan, Toulouse, France) for the kind gift of reagents.

## Additional information

### Funding

| Funder | Author |
| --- | --- |
| Schweizerische Nationalfonds zur Förderung der Wissenschaftlichen Forschung (Schweizerische Nationalfonds) | Didier Trono |
| European Research Council (ERC) | Didier Trono |

The funders had no role in study design, data collection and interpretation, or the decision to submit the work for publication.

### Author contributions

BR, Conception and design, Acquisition of data, Analysis and interpretation of data, Drafting or revising the article; SMJ, Acquisition of data, Analysis and interpretation of data, Drafting or revising the article; MC, IB, Analysis and interpretation of data, Drafting or revising the article; AK, Bioinformatical analysis, Analysis and interpretation of data, Drafting or revising the article; DT, Conception and design, Analysis and interpretation of data, Drafting or revising the article

### Ethics

Human subjects: Cord blood was obtained after approval of the project by the local ethics committee, commission cantonale d'ethique de la recherche sur l'etre humain, as protocol 146/10. All samples were obtained after simple oral informed consent of all participants by the obstetricians before the caesarian section, where necessary.

## Additional files

### Supplementary files

• Supplementary file 1. Localization of KAP1 peaks on HCMV TB40-E genome. GenBank: EF999921.1.

• Supplementary file 2. Primers used in this work and their sequences.

• Source code 1. Custom-made PERL program.

### Major dataset

The following dataset was generated:

| Author(s) | Year | Dataset title | Dataset ID and/or URL | Database, license, and accessibility information |
|---|---|---|---|---|
| Rauwel B, Jang SM, Cassano M, Kapopoulou A, Barde I, Trono D | 2015 | Release of Human Cytomegalovirus from latency by a KAP1/TRIM28 phosphorylation switch | http://www.ncbi.nlm.nih.gov/geo/query/acc.cgi?acc=GSE53271 | Publicly available at NCBI Gene Expression Omnibus (GSE53271). |

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
