## [Decision Letter]

Thank you for sending your work entitled “Release of Human Cytomegalovirus from latency by a KAP1/TRIM28 phosphorylation switch” for consideration at *eLife*. Your article has been favorably evaluated by Tadatsugu Taniguchi (Senior editor), a guest Reviewing editor, and two additional reviewers.

The Reviewing editor and the other reviewers discussed their comments before we reached this decision, and the Reviewing editor has assembled the following comments to help you prepare a revised submission.

*Reviewer 1*:

The manuscript by Rauwel et al. reports that the master co-repressor KAP1 is involved in regulating viral lytic gene expression during latent infection of undifferentiated myeloid cells by HCMV; KAP1 mediates recruitment of chromatin post-translational modifiers to the viral Major Immediate Early Promoter (MIEP) resulting in repression of the viral MIEP and the establishment of a latent transcription programme.

This is an interesting paper which presents data which goes some way to identifying at least one of the mechanisms by which the viral MIEP might be repressed during latent infection of myeloid cells with HCMV. However, in some cases the data could be presented more clearly, proper controls should be used and the data should be analysed in more depth.

The authors should address the following:

1) Figure 1 provides key data for the authors' hypothesis but conflate two very different analyses. Panel A and B analyse permissiveness of cells for infection (KAP1 is knocked-down, then cells are infected with HCMV); panel D analyses reactivation (cells are infected with HCMV, left to go latent and then KAP1 is knocked-down); it is unclear from the information in the figure legend how the cells were treated in panel C but my reading of the text suggests that panel C represents virus production from cells which were knocked-down for KAP1 prior to infection.

These different analyses are informative but should be separated clearly into panels which show analyses of permissiveness (KAP1 knock down then HCMV infection) and reactivation from latency (HCMV infection, then subsequent KAP1 knock-down). A key question then is whether cells that are reactivated from latency (HCMV infection, then subsequent KAP1 knock-down) actually produce virus in this model. This only appears to be shown, in panel C, for cells that have been made permissive (KAP1 knock down then HCMV infection). This is important, as delivery genome in the context of virions (with pp71 tegument activator etc.) is a very different condition to reactivation of latent genome (in which no such tegument proteins will be present).

The authors should carry out a comprehensive analysis of IE/E/late RNAs (as shown in panels 1 A/B) on cells that are induced to reactivate by KAP1 depletion (HCMV infection, then subsequent KAP1 knock-down) and also analyse these cells for de novo virus production (as shown in panel C). This is important as more data are accumulating to suggest that other hurdles, besides just induction of IE1/IE2, may be needed for full reactivation of infectious virus.

2) Figures 2 and 3. These analyses should be carried out side-by-side with latent myeloid cells that have been induced to reactivate by standard differentiation conditions (not compared to permissive fibroblasts, or to an laboratory isolate of virus that is missing a large amount of viral genome and is inefficient at establishing latency in CD34^+^ cells).

3) Figures 4 and 5. I am uncertain why the authors have reverted to using permissive fibroblasts for these analyses. They should be carried out in latent CD34^+^ which have been differentiated to reactivate latent HCMV genome (alternatively, latent monocytes differentiated to DCs could be used).

4) The authors must discuss their observations in the light of recently published data that mTOR inhibition does not prevent HCMV reactivation from dendritic cells (16) and that macrophages infected with HCMV, contrary to observations with fibroblasts, are sensitive to rapamycin (37). This reinforces the reason why some of the analyses in fibroblasts shown in Figures 4 and 5 should be carried out in differentiated myeloid cells.

Reviewer 2:

This paper seeks to examine a role for KAP1 in the control of HCMV gene expression during latency and reactivation. The premise is very interesting. The Trono group have performed some fascinating work looking to dissect the role of this protein in the context of endogenous retroviruses but in, this instance, the data does not convince.

A more minor issue is that the use of data in lytic data from infected MRC5 proves confusing and reduces the clarity of the report whereas data in the supplementary pertaining to latent cells should be in the main text to support the authors conclusions.

1) Figure 1 implies that KAP1 is important for the establishment of latency: KO of KAP1 function results in more IE gene expression. Yet the authors then go on in Figure 3—figure supplement 1 to study AD169, 'as a model virus that does not establish latency', and see that KAP1 binds this viral genome just as efficiently as TB40-E in CD34^+^ cells. The argument is that KAP1 exerts functions via recruitment of secondary modulators (HP1 etc.). This, to me, needs to be explored. For instance, a recent report from the Pari laboratory suggests HCMV encodes a viral RNA that recruits Polycomb repressor complex and H3K27meth marks at the MIEP. Why aren't HP1 etc. recruited? How does KAP1 effect these functions? This seems pivotal for understanding the role of KAP1. Ultimately, what is it about AD169 that negates any activity of KAP1? An alternative explanation could be that the data from analysing TB40-E and AD169 suggests that KAP1 binding to the viral genome is irrelevant and that any KAP1 effects observed in Figure 1 are indirect due to the multitude effects KAP1 KO would have on other cellular function which then impact on HCMV gene expression.

2) Which KAP1 (indirect) binding site do the authors believe is important for controlling the MIEP (given that it doesn't appear to bind to be recruited directly to the MIEP in Figure 2)? Some means to block KAP1 recruitment to the viral genome (via sequence mutagenesis) would provide more compelling evidence that KAP1 binding to the viral DNA drives heterochromatic formation at the MIEP. However, this would require identification of the KAP1 binding partner tethering it to the viral DNA. I appreciate the siRNA KO of KAP1 blocks heterochromatic formation at the MIEP but the AD169 observation requires explanation and again suggests that any KAP1 effects could be highly indirect and due to major changes in the cellular biology when knocking out such a key regulator. Again the AD169 data really argues against KAP1 binding being important for controlling latency since KAP1 is recruited to this genome perfectly well but does not provide the same level of control as it apparently does in TB40-E.

3) The authors make an important observation in their studies of CD34^+^ precursors (Figure 8). The chloroquinone expt (thought to be via KAP1) has only a minor effect on HCMV reactivation and you need subsequent NF-kB stimulation to make it more robust. But they never compare it to DCs directly they just make inference (even DCs given these additional treatments). This would be very interesting and provide a more complete set of data. That aside, is the prediction that if you were to KO KAP1 and see the effect observed in Figure 1 then the addition of chloroquinone (thought to be active via modulation of KAP1) to KAP1 KO cells should have no incremental impact on HCMV reactivation if the authors predictions are true i.e. if KAP1 mediated repression is deleted by siRNA then any effects of chloroquinone should be abrogated.

In summary, the link between KAP1, heterochromatic control of viral latency and the activation of mTOR pathways is all correlative. I think there has, at least, to be some appreciation that whilst pharmacological modulators of KAP1 could trigger reactivation the data does not state it is the activity of those inhibitors towards KAP1 which is mediating the effect. One would presume ATM promotes phosphorylation of a number of cellular proteins but that does not mean they are all involved in HCMV reactivation. The paper sets out to show that KAP1 is the regulator of HCMV latency and reactivation and becomes less and less convincing. If the premise of the paper was chloroquinone and ATM trigger HCMV reactivation and we aim to identify a mechanism it would be more compelling.

Reviewer 3:

The authors provide evidence that shKAP1 transduction can induce high levels of hCMV transcription and virus production in latently infected CD34^+^ cells, but not in permissively infected MRC-5 cells. They show by ChIP-Seq and ChIP-PCR that KAP1 binds to some locations of the HCMV genome in CD34^+^ cells. ChIP with KAP pS824-specific antibody suggests that phosphorylated KAP is enriched in MRC-5 cells, but not CD34^+^ cells. They show that chloroquine treatment induces HCMV reactivation from CD34^+^ cells, and that this is attenuated by treatment with an ATM inhibitor, but not an inhibitor of mTOR. They also show that addition of TNF augments chloroquine activation of HCMV transcription and viral production in CD34^+^ cells. The authors conclude that HCMV latency is regulated by ATM or mTOR phosphorylation of KAP1 S824.

This is an interesting paper, with many intriguing findings covering a great breadth of phenomena that control HCMV latency and reactivation. However, while there is evidence for each claim, the evidence is often minimal and without sufficient controls to rule out alternative explanations. Some of the experimental data (especially the imaging data) is lacking critical controls and quantification. Without deeper analysis of each of the many conclusions, the manuscript appears superficial and the conclusions are not strongly justified.

Specific comments:

1) Figure 2, ChIP-Seq: the statistics of the ChIP-Seq should be provided. How many total reads, how many reads mapped back to the CMV genome. Also, the peak calling algorithm is not entirely clear. What is the quality of the ChIP-Seq on cellular genome (this should be relatively simple to provide and will serve as quality control for the ChIP-Seq experiment). Also, the methods for normalization of the ChIP-Seq and ChIP-PCR (Figure 3) are not clearly described.

2) What sequence specific DNA binding proteins on the viral genome recruit KAP? Are there any characterized KRAB Zn finger proteins at each of these KAP bound sites?

3) Figure 3: While the absolute values of pS824 KAP binding in CD34 cells are low, they are at similar levels relative to the CD34 KAP binding observed for the MRC-5. In other words, the observed percentage of pS^824^KAP in CD34 (10:2) and MRC-5 (40:8) relative to total KAP are identical.

4) Figure 4 requires some statistical/quantitative analysis- how representative are these images?

5) Figure 5, mTOR ChIP assays: it is untypical for a kinase-like mTOR to remain associated with its substrate and viral chromatin. Is there any evidence that mTOR remains stably associated with KAP by coIP experiment?

6) Figure 5: is not very convincing or clear as to what is colocalized and where. It is also not clear what stage of infection is presented for each virus, and how typical this is of CMV infection.

7) What is the timing of mTOR recruitment to the HCMV genome during reactivation? If it is recruited to the replication compartments, than it must be recruited very late in the process after early gene transcription. How does this fit with the model the mTOR may be regulating KAP control of viral transcription and reactivation.

8) The effects of chloroquine on HCMV viral production is impressive, but chloroquine has many pleiotropic effects on the cell and it is not clear that KAP is a major regulator under these conditions. One way to demonstrate the importance of the KAP pS^824^ in control of HCMV would be to re-introduce the KAP wt or KAP S^824^A into shKAP depleted cells and assay HCMV reactivation. A mutation like S^824^A can be assayed for its ability to block KAP phosphorylation and viral reactivation.

9) The different effects of TORIN and KU on HCMV are interesting, but may not fit well with the overall model of a KAP1 phospho-switch. The effects of these inhibitors on KAP pS^824^ needs to be assessed by Western blot during normal infection and reactivation induced by chloroquine.

---

## [Author Response]

Reviewer 1:

*[…] The authors should address the following*:

*1)*
Figure 1
*provides key data for the authors' hypothesis but conflate two very different analyses. Panel A and B analyse permissiveness of cells for infection (KAP1 is knocked-down, then cells are infected with HCMV); panel D analyses reactivation (cells are infected with HCMV, left to go latent and then KAP1 is knocked-down); it is unclear from the information in the figure legend how the cells were treated in panel C but my reading of the text suggests that panel C represents virus production from cells which were knocked-down for KAP1 prior to infection*.

The Figure 1 legend was complemented, we effectively forgot to mention that panel C represents virus production from cells depleted for KAP1 prior to infection and that it completes results exhibited in panel B.

*These different analyses are informative but should be separated clearly into panels which show analyses of permissiveness (KAP1 knock down then HCMV infection) and reactivation from latency (HCMV infection, then subsequent KAP1 knock-down). A key question then is whether cells that are reactivated from latency (HCMV infection, then subsequent KAP1 knock-down) actually produce virus in this model. This only appears to be shown, in panel C, for cells that have been made permissive (KAP1 knock down then HCMV infection). This is important, as delivery genome in the context of virions (with pp71 tegument activator etc.) is a very different condition to reactivation of latent genome (in which no such tegument proteins will be present)*.

*The authors should carry out a comprehensive analysis of IE/E/late RNAs (as shown in panels 1 A/B) on cells that are induced to reactivate by KAP1 depletion (HCMV infection, then subsequent KAP1 knock-down) and also analyse these cells for de novo virus production (as shown in panel C). This is important as more data are accumulating to suggest that other hurdles, besides just induction of IE1/IE2, may be needed for full reactivation of infectious virus*.

The new Figure 1 provides a complete analysis of IE/E/L HCMV RNA levels in cells that are induced to reactivate the virus by KAP1 depletion as requested. It also demonstrates that viral activation, which can only be expected to be partial owing to the persistence of some KAP1 in the cells, can be boosted by combining the KAP1 knockdown with TNF-α mediated- NF-κB activation.

*2)*
Figures 2 and 3*. These analyses should be carried out side-by-side with latent myeloid cells that have been induced to reactivate by standard differentiation conditions (not compared to permissive fibroblasts, or to an laboratory isolate of virus that is missing a large amount of viral genome and is inefficient at establishing latency in CD34*^*+*^
*cells)*.

We had chosen the MRC-5 model to provide a clear comparison between latency and full-blown viral replication. Now we followed the reviewer’s excellent suggestion to monitor HCMV reactivation when latently infected CD34 cells are induced to differentiate into mature dendritic cells (mDC). We thus performed KAP1-, pS^824^KAP-, SetDB1-, H3K9Me3- and HP1α-specific ChIPs in infected CD34^+^-derived- mDCs (New Figure 3 and Figure 2—figure supplement 1) and transferred the corresponding MRC-5 data to Figure 3—figure supplement 1. The new data nicely confirm that, in HCMV-replicating mDCs, it is the pS^824^ of KAP1 that is associated with the viral genome, which is correspondingly devoid of SETDB1 and H3K9me3.

*3)*
Figures 4 and 5*. I am uncertain why the authors have reverted to using permissive fibroblasts for these analyses. They should be carried out in latent CD34*^*+*^
*which have been differentiated to reactivate latent HCMV genome (alternatively, latent monocytes differentiated to DCs could be used)*.

We performed the IF studies in fibroblast because these cells are far more amenable to this type of analysis than mDCs due both to their higher rates of infection and to their adherent status. Since mDCs and MRC5 cells are comparable in terms of the HCMV recruitment of KAP1 subspecies, SETDB1 and H3K9me3 (see new Figure 3), we think that this is fair.

*4) The authors must discuss their observations in the light of recently published data that mTOR inhibition does not prevent HCMV reactivation from dendritic cells (*[16]*) and that macrophages infected with HCMV, contrary to observations with fibroblasts, are sensitive to rapamycin (*[37]*). This reinforces the reason why some of the analyses in fibroblasts shown in*
Figures 4 and 5
*should be carried out in differentiated myeloid cells*.

The Discussion was modified to include a sentence on the recently published Glover et al.’s paper. The fact that rapamycin has no effect on in vitro HCMV reactivation in dendritic cells could be explained by the addition of the drug to immature cells, where we could already observe low levels of viral expression, suggesting that KAP1 is already at least partly phosphorylated in this setting. Which may be different in monocytes/macrophages, where mTOR seems accordingly be more important. Exploring these other settings, both from an epigenetic standpoint and regarding possible effects of mTOR on later steps of viral replication, is beyond the scope of this manuscript.

Reviewer 2:

*This paper seeks to examine a role for KAP1 in the control of HCMV gene expression during latency and reactivation. The premise is very interesting. The Trono group have performed some fascinating work looking to dissect the role of this protein in the context of endogenous retroviruses but in, this instance, the data does not convince*.

*A more minor issue is that the use of data in lytic data from infected MRC5 proves confusing and reduces the clarity of the report whereas data in the supplementary pertaining to latent cells should be in the main text to support the authors conclusions*.

*1)*
Figure 1
*implies that KAP1 is important for the establishment of latency: KO of KAP1 function results in more IE gene expression. Yet the authors then go on in*
Figure 3—figure supplement 1
*to study AD169, 'as a model virus that does not establish latency', and see that KAP1 binds this viral genome just as efficiently as TB40-E in CD34*^*+*^
*cells. The argument is that KAP1 exerts functions via recruitment of secondary modulators (HP1 etc.). This, to me, needs to be explored. For instance, a recent report from the Pari laboratory suggests HCMV encodes a viral RNA that recruits Polycomb repressor complex and H3K27meth marks at the MIEP. Why aren't HP1 etc. recruited? How does KAP1 effect these functions? This seems pivotal for understanding the role of KAP1. Ultimately, what is it about AD169 that negates any activity of KAP1? An alternative explanation could be that the data from analysing TB40-E and AD169 suggests that KAP1 binding to the viral genome is irrelevant and that any KAP1 effects observed in*
Figure 1
*are indirect due to the multitude effects KAP1 KO would have on other cellular function which then impact on HCMV gene expression*.

Our data, whether illustrated in the first version of our manuscript or added since then (see new Figures 1 and 3), fully support a model whereby KAP1 is associated with the HCMV genome in both latently and productively infected cells, but differentially impacts on viral gene expression according to its S^824^ phosphorylation status, which is explained by the inhibitory effect of this post-translational modification on the secondary recruitment of HP1 and SETDB1, hence on the deposition of repressive marks including H3K9me3. Why viral strains such as AD169 fail to enter latency in CD34 cells is an interesting issue, which is discussed in our manuscript, but the experimental exploration of which is beyond the scope of this study.

*2) Which KAP1 (indirect) binding site do the authors believe is important for controlling the MIEP (given that it doesn't appear to bind to be recruited directly to the MIEP in*
Figure 2*)? Some means to block KAP1 recruitment to the viral genome (via sequence mutagenesis) would provide more compelling evidence that KAP1 binding to the viral DNA drives heterochromatic formation at the MIEP. However, this would require identification of the KAP1 binding partner tethering it to the viral DNA. I appreciate the siRNA KO of KAP1 blocks heterochromatic formation at the MIEP but the AD169 observation requires explanation and again suggests that any KAP1 effects could be highly indirect and due to major changes in the cellular biology when knocking out such a key regulator. Again the AD169 data really argues against KAP1 binding being important for controlling latency since KAP1 is recruited to this genome perfectly well but does not provide the same level of control as it apparently does in TB40-E*.

We previously demonstrated that heterochromatin can spread tens of kilobases away from a primary genomic KAP1 recruiting site ([20], reference 31 herein). Here, we demonstrate that it also true in the HCMV episome, with both H3K9me3 and HP1 extending to the MIEP in spite of the absence of direct KAP1 binding at this locus.

Blocking KAP1 recruitment via mutagenesis would require the fine mapping of KAP1 recruiting site and their deletion or mutation, which for close to 30 such sites detected on the HCMV genome is not only daunting but unlikely to be possible without profoundly perturbing other features essential to the virus.

For AD169-related comments, see above.

*3) The authors make an important observation in their studies of CD34*^*+*^
*precursors (*Figure 8*). The chloroquinone expt (thought to be via KAP1) has only a minor effect on HCMV reactivation and you need subsequent NF-kB stimulation to make it more robust. But they never compare it to DCs directly they just make inference (even DCs given these additional treatments). This would be very interesting and provide a more complete set of data. That aside, is the prediction that if you were to KO KAP1 and see the effect observed in*
Figure 1
*then the addition of chloroquinone (thought to be active via modulation of KAP1) to KAP1 KO cells should have no incremental impact on HCMV reactivation if the authors predictions are true i.e. if KAP1 mediated repression is deleted by siRNA then any effects of chloroquinone should be abrogated*.

In response to the reviewer’s excellent suggestion, we performed pharmacological reactivation on KAP1 KD CD34^+^ cells whether or not complemented with wild-type KAP1 or with the non-phosphorylatable KAP1 S824A mutant (new Figure 6—figure supplement 6). We observed that chloroquine had no additional impact on the levels of HCMV reactivation induced by KAP1 depletion, and that it could reactivate the virus in KD cells rescued with wild type but not S^824^A KAP1. It strongly supports a model whereby chloroquine de-represses HCMV gene expression through KAP1 phosphorylation, as expected from its ability to activate ATM.

*In summary, the link between KAP1, heterochromatic control of viral latency and the activation of mTOR pathways is all correlative. I think there has, at least, to be some appreciation that whilst pharmacological modulators of KAP1 could trigger reactivation the data does not state it is the activity of those inhibitors towards KAP1 which is mediating the effect. One would presume ATM promotes phosphorylation of a number of cellular proteins but that does not mean they are all involved in HCMV reactivation. The paper sets out to show that KAP1 is the regulator of HCMV latency and reactivation and becomes less and less convincing. If the premise of the paper was chloroquinone and ATM trigger HCMV reactivation and we aim to identify a mechanism it would be more compelling*.

Reviewer 3:

*[…] This is an interesting paper, with many intriguing findings covering a great breadth of phenomena that control HCMV latency and reactivation. However, while there is evidence for each claim, the evidence is often minimal and without sufficient controls to rule out alternative explanations. Some of the experimental data (especially the imaging data) is lacking critical controls and quantification. Without deeper analysis of each of the many conclusions, the manuscript appears superficial and the conclusions are not strongly justified*.

*Specific comments*:

*1)*
Figure 2*, ChIP-Seq: the statistics of the ChIP-Seq should be provided. How many total reads, how many reads mapped back to the CMV genome. Also, the peak calling algorithm is not entirely clear. What is the quality of the ChIP-Seq on cellular genome (this should be relatively simple to provide and will serve as quality control for the ChIP-Seq experiment). Also, the methods for normalization of the ChIP-Seq and ChIP-PCR (*Figure 3*) are not clearly described*.

All statistics and Datasets of the ChIP-Seq will be available on Geo as described in Material and methods. We had 163 Million total reads and mapped 135 Million of them on the human genome. Besides this, 400,000 reads corresponded to the HCMV genome. As the CMV-specific signal was broad, the MACS software was not appropriate for peak calling, requesting the development of a special algorithm. All positive, poorly enriched and negative regions were also analyzed by qPCR, which confirmed the results of the ChIP-Seq. The method of ChIP-PCR normalization is explained in the Material and methods section. The Quality control for the ChIP-seq is provided in Figure 9, together with its Total Input Quality control in Figure 10:

Author response image 1.**DOI:**
http://dx.doi.org/10.7554/eLife.06068.036

Author response image 2.**DOI:**
http://dx.doi.org/10.7554/eLife.06068.037

Examples of positive peaks on the cellular genome as additional quality controls for the experiment are shown in Figure 11, Figure 12, and Figure 13:

Author response image 3.**DOI:**
http://dx.doi.org/10.7554/eLife.06068.038

Author response image 4.**DOI:**
http://dx.doi.org/10.7554/eLife.06068.039

Author response image 5.**DOI:**
http://dx.doi.org/10.7554/eLife.06068.040

2) What sequence specific DNA binding proteins on the viral genome recruit KAP? Are there any characterized KRAB Zn finger proteins at each of these KAP bound sites?

Good questions… At this stage, we do not know. It may indeed be that KAP1 is recruited by KRAB-ZNF proteins. This is consistent with the failure of an RBCC-deleted KAP1 mutant to associate with the HCMV genome, as the RBCC domain of KAP1 is what recognizes KRAB-ZNFs, but this domain can also mediate other types of protein-protein interactions.

*3)*
Figure 3*: While the absolute values of pS824 KAP binding in CD34 cells are low, they are at similar levels relative to the CD34 KAP binding observed for the MRC-5. In other words, the observed percentage of pS*^*824*^*KAP in CD34 (10:2) and MRC-5 (40:8) relative to total KAP are identical*.

It is impossible to compare levels of enrichment obtained by performing ChIP with 2 different antibodies in 2 different cell models. Nevertheless, we observed that with KAP1-specific antibody, which is able to precipitate the phosphorylated and non-phosphorylated forms, there are regions of the HCMV genome that score negative (depicted as HCMV Neg1 and HCMV Neg 2). The pS824 KAP1-specific ChIPs gave positive results on the 28 peaks scored positive with the KAP1-specific antibody, but also did not detect a signal at these neg1 and neg2 regions in mDC or MRC-5 fibroblasts, but scored globally negative in latently infected CD34^+^, which provides the necessary controls.

*4)*
Figure 4
*requires some statistical/quantitative analysis- how representative are these images?*

Even though only 3 or 4 cells are illustrated on the picture, we analyzed a large number of them (several tens) to make sure that we were not misled by non-representative occurrences.

*5)*
Figure 5*, mTOR ChIP assays: it is untypical for a kinase-like mTOR to remain associated with its substrate and viral chromatin. Is there any evidence that mTOR remains stably associated with KAP by coIP experiment?*

The genomic recruitment of a kinase was previously described in the context of MSK1-HP1γ complexes (Harouz H, et al., Embo J., 2014, Nov 18;33(22):2606-22) and also with KAP1-MSK1 (our lab : Singh K. et al., Genes and Dev., 2015, in press). We do not have evidence that mTOR is stably associated with KAP1, and we do not expect it to be the case: instead, we think that the kinase is only recruited to HCMV-bound KAP1 molecules, as suggested by the finding that the kinase is still able to repress a genomic target in HCMV-infected cells. Of note, the ChIP protocol most likely “fixes” an interaction that might otherwise be not so stable.

*6)*
Figure 5*: is not very convincing or clear as to what is colocalized and where. It is also not clear what stage of infection is presented for each virus, and how typical this is of CMV infection*.

We do not claim colocalization of IE and mTOR. We are just illustrating (with help of white arrows) that, in infected cells (expressing IE) some mTOR-positive foci can be detected in the nucleus whereas in uninfected the mTOR signal diffuse and cytoplasmic.

*7) What is the timing of mTOR recruitment to the HCMV genome during reactivation? If it is recruited to the replication compartments, than it must be recruited very late in the process after early gene transcription. How does this fit with the model the mTOR may be regulating KAP control of viral transcription and reactivation*.

We agree with the reviewer that, according to our findings, it is the recruitment of mTOR or some other KAP1-targeting kinase at least to some HCMV-bound KAP1 molecules that jumpstarts viral gene expression by relieving heterochromatic marks. As such, it cannot be defined as happening in “viral replication compartments”, because it precedes viral replication. We modified the corresponding statement in the discussion. Fine monitoring of the timing of mTOR recruitment in a system where HCMV naturally exits from latency to induce productive infection would be worthwhile, but it is not straightforward, hence beyond the scope of the present study.

*8) The effects of chloroquine on HCMV viral production is impressive, but chloroquine has many pleiotropic effects on the cell and it is not clear that KAP is a major regulator under these conditions. One way to demonstrate the importance of the KAP pS*^*824*^
*in control of HCMV would be to re-introduce the KAP wt or KAP S*^*824*^*A into shKAP depleted cells and assay HCMV reactivation. A mutation like S*^*824*^*A can be assayed for its ability to block KAP phosphorylation and viral reactivation*.

*Very good suggestions, which helped strengthen out model through additional experiments illustrated in new*
Figure 6—figure supplement 6
*as explained in our response to reviewer 1, point 3*.

*9) The different effects of TORIN and KU on HCMV are interesting, but may not fit well with the overall model of a KAP1 phospho-switch. The effects of these inhibitors on KAP pS*^*824*^
*needs to be assessed by Western blot during normal infection and reactivation induced by chloroquine*.

The effect of Torin and KU on KAP1 phosphorylation was documented by IF in the MRC-5 model, as depicted in Figure 4 and Figure 4—figure supplement 1. The corresponding antibody works poorly for Western blot, and monitoring KAP1 phosphorylation during viral reactivation in low numbers of HCMV-infected cells is beyond our current technical abilities.